# TRANSLUTION: UNIFYING SELF-ATTENTION AND CONVOLUTION FOR ADAPTIVE AND RELATIVE MODELING

## ABSTRACT

When modeling a given type of data, we consider it to involve two key aspects: 1) identifying relevant elements (*e.g.*, image pixels or textual words) to a central element, as in a convolutional receptive field, or to a query element, as in self-attention, and 2) encoding these tokens effectively. Self-attention can adaptively identify these elements but relies on absolute positional embedding for structural representation learning. In contrast, convolution encodes elements in a relative manner, yet their fixed kernel size limits their ability to adaptively select the relevant elements. In this paper, we introduce Translution, an operation that unifies the adaptive identification capability of self-attention and the relative encoding advantage of convolution. However, this integration leads to a substantial increase in the number of parameters, exceeding most currently available computational resources. Therefore, we propose a lightweight low-rank variant of Translution, named LoR-Translution. Experiments on computer vision and natural language processing tasks show that Translution (including LoR-Translution) achieves superior accuracy compared to self-attention. The code has been included in the supplementary materials and will be released soon.

## 1 INTRODUCTION

Recent evidence suggests that directly scaling up deep neural networks, particularly Transformers (Vaswani et al., 2017; Radford et al., 2018; Devlin et al., 2019; Dosovitskiy et al., 2021), with additional data and parameters is encountering diminishing returns. Leading Artificial Intelligence (AI) labs have similarly noted slower-than-anticipated improvements in next-generation models, despite extensive training efforts. Given the saturation of available data and limitations imposed by current scaling laws, it is crucial now to reflect on past successes and pursue the design of innovative neural networks to sustain future progress in deep learning.

When employing deep neural networks to model a specific type of data, the process can be decomposed into two key aspects: 1) identifying relevant data elements and 2) encoding these elements into effective representations. When using convolutional neural networks (LeCun et al., 1998; Krizhevsky et al., 2012; Simonyan & Zisserman, 2015; Szegedy et al., 2015; He et al., 2016) to process images, the basic element is pixel. When using Transformers, the element is word for natural language processing and patch for visual tasks.

### 1.1 IDENTIFICATION OF RELEVANT ELEMENTS

In convolution, as shown in Figure 1 (a), the relevant element identification step is handled by convolutional filters (kernels) with a fixed local receptive field. This fixed kernel defines a neighborhood that is considered relevant to the center. For visual data like images, such local focus is often effective because spatially adjacent pixels tend to be related (*e.g.*, forming parts of the same object). However, the rigid nature of a fixed-size kernel makes convolution inevitably cover irrelevant pixels, especially near object boundaries or in background areas that fall inside the window.

In contrast, as shown in Figure 1 (b), self-attention (Vaswani et al., 2017) can adaptively identify relevant regions. Instead of being limited to a predetermined locality, it allows the model to dy-

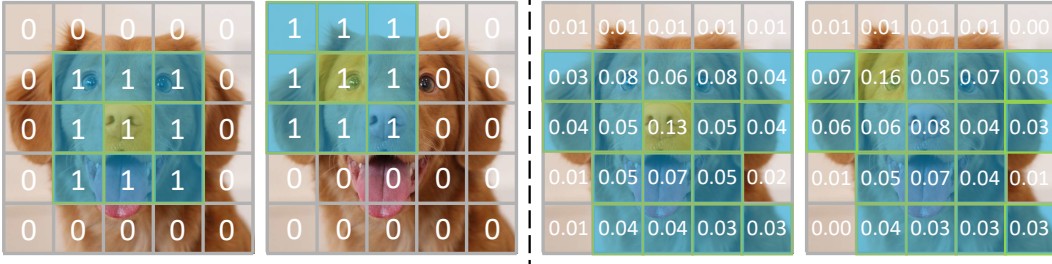

|          | (a) Convolution |          | (b) Self-attention |

Figure 1: Difference between convolution and self-attention in identifying relevant elements (blue patches) for the kernel center or query element (yellow patch). Here, convolution is assumed to operate on image patches. 1) Convolution utilizes a fixed kernel size to define a neighborhood of elements considered relevant, inevitably including some irrelevant regions, particularly near object boundaries or within background areas inside the window. The fixed receptive field in convolution can be interpreted as a special case of attention, where the attention score is set to 1 within the receptive field and 0 outside it. 2) Self-attention adaptively identifies relevant elements by assigning greater attention scores to areas with higher relevance, thereby mitigating the inclusion of noisy or irrelevant information.

namically attend to relevant regions. This means that self-attention can focus on important features regardless of their physical distance. This capability provides greater flexibility compared to the convolution's fixed receptive field.

## 1.2 ENCODING OF RELEVANT ELEMENTS

When it comes to encoding the structure from these relevant elements, convolution and self-attention employ different strategies. As shown in Figure 2 (a), a convolutional kernel learns distinct parameters $\{\boldsymbol{W}_{\delta_x,\delta_y}\}$ for each relative direction and distance within its receptive field. In other words, the filter has separate parameters $\boldsymbol{W}_{\delta_x,\delta_y}$ for each offset $\delta_x, \delta_y$ from the center. This design enables convolution to encode local structure relatively — capturing orientation and distance relationships.

In contrast, as shown in Figure 2 (b), self-attention uses three shared sets of parameters $\boldsymbol{W}^q$, $\boldsymbol{W}^k$ and $\boldsymbol{W}^v$ to process inputs for all positions. Consequently, the query, key and value of self-attention do not encode whether one patch is to the left or right of another. To introduce positional information, Transformer incorporates absolute positional embeddings into the input features at the outset. Although these embeddings enable Transformer to infer order or spatial relationships, they introduce noise into each token's representation. The absolute position information becomes part of the input features. Consequently, when the same object moves to a different location, Transformer may struggle to recognize it.

## 1.3 UNIFICATION OF CONVOLUTION AND TRANSFORMER

In summary, convolution encodes structure through fixed local filters with position-specific weights, whereas self-attention relies on adaptive global attention and requires absolute positional encoding to capture order or spatial structures.

In this paper, we introduce Translution, a new type of operation that unifies the adaptive identification capability of self-attention with the relative encoding advantage of convolution. Specifically, Translution employs a convolution-style approach that assigns separate parameters (matrices) to each distance and direction when computing the query, key and value. This design enables Translution to effectively encode relative structures.

However, this unification leads to a significant increase in the number of parameters and exceeds most currently available computational resources. Therefore, we propose a lightweight variant of Translution, named LoR-Translution, which significantly reduces the number of parameters. This variant achieves lower accuracy than the "ideal" (original) Translution but better accuracy than self-attention.

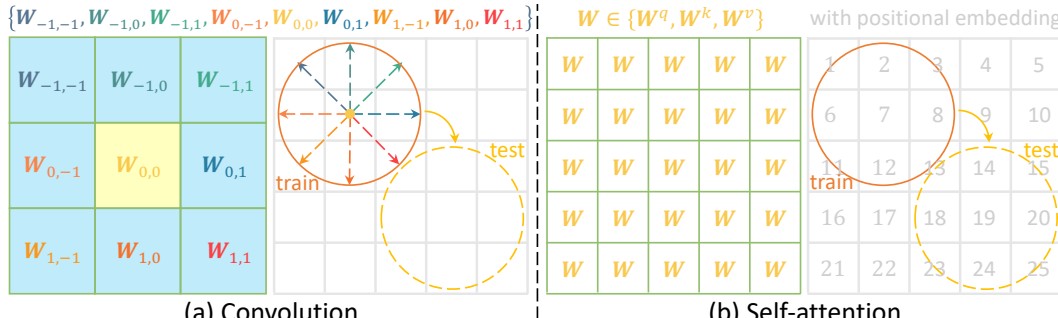

Figure 2: Difference between convolution and self-attention in encoding relevant elements: consider the scenario where convolution and self-attention are capturing the structure of a circle. 1) Convolution learns separate parameters $\{W_{\delta_x,\delta_y}\}$ for each offset, where $\delta_x, \delta_y \in [-1, 1]$, from the kernel center, allowing it to effectively encode relative local structures. Thus, when the circle appears in a different location, it is still readily recognized due to this relative awareness. 2) Self-attention incorporates absolute position into each token's representation and uses position-irrelevant parameters $W \in \{W^q, W^k, W^v\}$ across all tokens for computing query, key and value, respectively. While this method facilitates general processing, the inclusion of absolute positional embeddings makes it more challenging to recognize the circle when it is moved to a different location.

As a fundamental operation, we investigate whether Translution can outperform self-attention. We conduct experiments on two widely-used Transformer architectures: Vision Transformer (ViT) (Dosovitskiy et al., 2021) for computer vision tasks and Generative Pre-trained Transformer (GPT) (Radford et al., 2018; 2019; Brown et al., 2020) for natural language processing tasks. Experiments demonstrate that Translution and LoR-Translution surpass self-attention in terms of accuracy.

## 2 RELATED WORK

Transformer (Vaswani et al., 2017; Radford et al., 2018; Devlin et al., 2019; Dosovitskiy et al., 2021; Liu et al., 2021; Touvron et al., 2021) eschews recurrence (as used in recurrent neural networks) and kernel size (as used in convolutional neural networks), instead employing self-attention for relevant region identification. Because it has no built-in notion of order, Transformer incorporates explicit absolute positional embeddings into token embeddings, enabling the model to utilize sequence order. Subsequent work has explored "relative attention" (Shaw et al., 2018; Huang et al., 2019; Parmar et al., 2019; Dai et al., 2019; Tsai et al., 2019; Raffel et al., 2020; Dai et al., 2021), which integrates relative position information into self-attention. They can be categorized into three families: *1) Relative positional vector.* Shaw *et al.*enhanced Transformer for language modeling by adding learnable relative positional vectors into the key and value computations, respectively (Shaw et al., 2018). BoTNet (Srinivas et al., 2021) and HaloNet (Vaswani et al., 2021) extended this approach to two dimensions for image processing by adding learnable relative positional vectors into key. *2) Relative positional scalar.* Swin Transformer (Liu et al., 2021), CoAtNet (Dai et al., 2021), and ConViT d'Ascoli et al. (2021) incorporate a learnable relative positional bias (a scalar) into the attention score. In these methods, the original self-attention can be regarded as content attention, which measures relationships from the token-feature perspective, while the additional relative positional bias can be regarded as position attention, which measures relationships from the token-position perspective. *3) Rotary position embedding.* RoFormer (Su et al., 2024) introduces a rotary position embedding mechanism, which encodes relative positional information by applying a rotation operation in the Query and Key representation space. Unlike these existing methods, Translution employs a convolution-style approach that uses relative positional matrices for query, key and value computation. Section D provides a formal comparison of these methods.

Convolutional neural networks (LeCun et al., 1998; Krizhevsky et al., 2012; Simonyan & Zisserman, 2015; Szegedy et al., 2015; He et al., 2016) have been the backbone of deep learning for years. By using small, shared kernels and pooling, convolutional neural networks efficiently capture local patterns. Recent architectural developments integrate self-attention with convolution. For instance, X-volution Chen et al. (2021) introduces a theoretically grounded self-attention approximation that unifies local and global interactions within a multi-branch module, which can be structurally re-

parameterized into a single convolution operator for efficient deployment. Conformer (Gulati et al., 2020) combines convolution layers and self-attention layers to capture both local and global dependencies in audio sequences. Similarly, CeiT (Yuan et al., 2021) uses convolutions to extract low-level features and self-attention to model long-range dependencies. Restormer Zamir et al. (2022) integrates Transformer blocks with two key components, *i.e.*, Multi-DConv Head Transposed Attention and the Gated DConv Feed-Forward Network, forming an efficient architecture tailored for high-resolution image restoration. It can captures long-range pixel dependencies in the spirit of Transformers, while retaining the scalability and practicality of CNNs for large images. Unlike most existing methods, Translution operates at the basic module or layer level, blending the advantages of self-attention and convolution into a unified fundamental operation. It preserves the same input–output interface as self-attention while augmenting it with relative positional encoding capabilities on top of self-attention's adaptive modeling. It is fully compatible with existing self-attention implementations.

## 3 PRELIMINARY: CONVOLUTION AND SELF-ATTENTION

### 3.1 CONVOLUTION

Suppose $\boldsymbol{f}_{x,y} \in \mathbb{R}^{1 \times C}$ denotes the feature or representation at location $(x, y)$ in an image of height $H$ and width $W$, where $C$ is the number of the input feature channels. Convolution is designed to capture the local structure centered at $(x, y)$ with a fixed kernel size $h \times w$,

$$\boldsymbol{f}'_{x,y} = \sum_{\delta_x=-\lfloor h/2 \rfloor}^{\lfloor h/2 \rfloor} \sum_{\delta_y=-\lfloor w/2 \rfloor}^{\lfloor w/2 \rfloor} \boldsymbol{f}_{x+\delta_x,y+\delta_y} \cdot \boldsymbol{W}_{\delta_x,\delta_y},$$

where $\boldsymbol{W}_{\delta_x,\delta_y} \in \mathbb{R}^{C \times C'}$ denotes the learnable parameters corresponding to the displacement $(\delta_x, \delta_y)$, $C'$ indicates the output feature dimension, and $\cdot$ denotes matrix multiplication. By assigning a set of parameters for each offset within the receptive field, convolution is able to discern direction and distance, and capture the local structure relatively. This means that when the absolute location of an object changes, it can still capture the same structure. However, convolution employs a rigid method to identify relevant regions, *i.e.*, using a fixed-size window, making it inevitably include irrelevant pixels or regions — particularly near object boundaries or in background areas within the window.

### 3.2 SELF-ATTENTION

Suppose $\boldsymbol{x}_i \in \mathbb{R}^{1 \times C}$ represents the feature or representation of the $i$-th patch at location $(x_i, y_i)$. Transformer (Vaswani et al., 2017) first incorporates the embedding of absolute position into the input $\boldsymbol{x}_i$, as follows,

$$\text{input positional embedding:} \quad \boldsymbol{f}_i = \boldsymbol{x}_i + \text{Embed}(x_i, y_i).$$

Then, self-attention performs two separate linear projections on the feature to generate query $\boldsymbol{q}_i \in \mathbb{R}^{1 \times C'}$ and key $\boldsymbol{k}_j \in \mathbb{R}^{1 \times C'}$, where $C'$ is the dimension for query or key,

$$\text{query encoding:} \quad \boldsymbol{q}_i = \boldsymbol{f}_i \cdot \boldsymbol{W}^q,$$
$$\text{key encoding:} \quad \boldsymbol{k}_j = \boldsymbol{f}_j \cdot \boldsymbol{W}^k,$$

where $\boldsymbol{W}^q / \boldsymbol{W}^k \in \mathbb{R}^{C \times C'}$. Subsequently, scaled dot-product attention is computed for each query, and a softmax function is applied to normalize the attention weights for a query across all positions,

$$\text{attention:} \quad a_{i,j} = \frac{\boldsymbol{q}_i \cdot \boldsymbol{k}_j^T}{\sqrt{C'}}, \quad \alpha_{i,j} = \frac{e^{a_{i,j}}}{\sum_{n=1}^{N} e^{a_{i,n}}},$$

where $N = H \times W$. Next, self-attention conducts another linear projection on the input feature to generate value $\boldsymbol{v}_i \in \mathbb{R}^{1 \times C'}$, as follows,

$$\text{value encoding:} \quad \boldsymbol{v}_j = \boldsymbol{f}_j \cdot \boldsymbol{W}^v,$$

where $\boldsymbol{W}_v \in \mathbb{R}^{C \times C'}$. Finally, the output is computed as a weighted sum of the values, *i.e.*,

$$\text{weighted sum:} \quad \boldsymbol{f}_i' = \sum_{j=1}^{N} \alpha_{i,j} \times \boldsymbol{v}_j,$$

where $\boldsymbol{f}_i' \in \mathbb{R}^{1 \times C'}$. In this way, self-attention can adaptively search for related regions, providing greater flexibility than methods that use local fixed-size windows. However, unlike convolution, which learns a feature encoding for every direction and distance, self-attention does not encode the structure in a relative manner.

## 3.3 TRANSLUTION

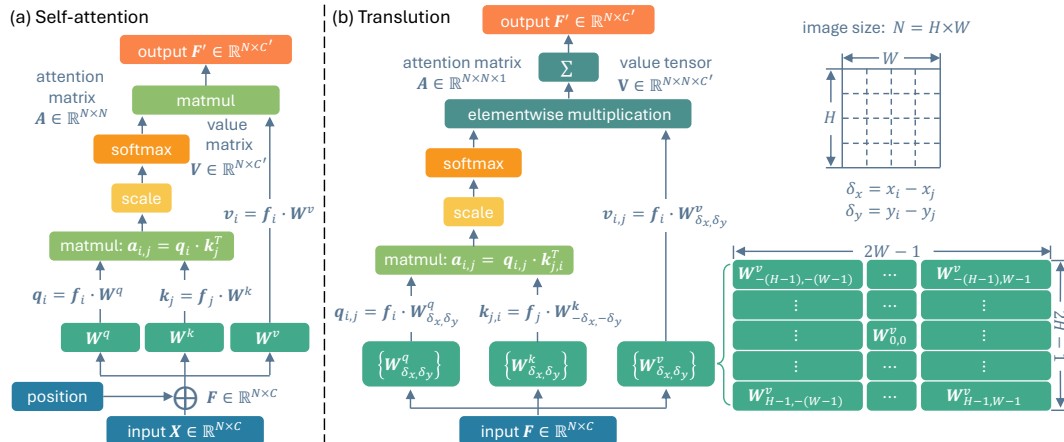

Figure 3: Comparison of self-attention and Translution. 1) Self-attention employs three shared sets of weights, *i.e.*, $\boldsymbol{W}^q$, $\boldsymbol{W}^k$, and $\boldsymbol{W}^v$, across all patches to compute query, key, and value, respectively. 2) Translution uses separate parameters for each offset (direction and distance), *i.e.*, $\{\boldsymbol{W}_{\delta_x,\delta_y}^q\}$, $\{\boldsymbol{W}_{\delta_x,\delta_y}^k\}$ and $\{\boldsymbol{W}_{\delta_x,\delta_y}^v\}$, to encode relative structures.

Translution is designed to integrate the adaptive related region identification capabilities of self-attention with the relative encoding strengths of convolution. Specifically, as shown in Figure 3, Translution employs a convolution-style formulation by assigning different parameters to compute query, key, and value, respectively, as follows:

$$\text{Translution} \begin{cases} \text{relative query encoding:} \ \boldsymbol{q}_{i,j} = \boldsymbol{f}_i \cdot \boldsymbol{W}_{\delta_x,\delta_y}^q, \ \delta_x = x_i - x_j, \ \delta_y = y_i - y_j, \\[2mm] \text{relative key encoding:} \ \boldsymbol{k}_{j,i} = \boldsymbol{f}_j \cdot \boldsymbol{W}_{-\delta_x,-\delta_y}^k, \\[2mm] \text{relative attention:} \ a_{i,j} = \dfrac{\boldsymbol{q}_{i,j} \cdot \boldsymbol{k}_{j,i}^T}{\sqrt{C'}}, \ \alpha_{i,j} = \dfrac{e^{a_{i,j}}}{\sum_{n=1}^{N} e^{a_{i,n}}}, \\[2mm] \text{relative value encoding:} \ \boldsymbol{v}_{i,j} = \boldsymbol{f}_j \cdot \boldsymbol{W}_{\delta_x,\delta_y}^v, \\[2mm] \text{weighted sum:} \ \boldsymbol{f}_i' = \sum_{j=1}^{N} \alpha_{i,j} \times \boldsymbol{v}_{i,j}, \end{cases} \quad (1)$$

where $\boldsymbol{W}_{\delta_x,\delta_y}^q / \boldsymbol{W}_{\delta_x,\delta_y}^k / \boldsymbol{W}_{\delta_x,\delta_y}^v \in \mathbb{R}^{C \times C'}$, represent the learnable parameter matrices for the query, key, and value corresponding to the displacement $(\delta_x, \delta_y)$.

*Translution unifies convolution and self-attention.*

The fixed receptive field in convolution can be interpreted as a special case of attention, where the attention score is set to 1 within the receptive field and 0 outside it, as shown in Figure 2. The weights $\boldsymbol{W}^q$, $\boldsymbol{W}^k$, and $\boldsymbol{W}^v$ in self-attention serve as shared linear projections that are uniformly applied across all spatial directions and distances. Consequently, Translution integrates the functionalities

of convolution and self-attention, as follows,

$$\text{Convolution:} \quad \boldsymbol{f}'_i = \sum_{j=1}^{N} \alpha_{i,j} \times \boldsymbol{f}_j \cdot \boldsymbol{W}_{\delta_x, \delta_y}, \quad \text{where } \alpha_{i,j} = \begin{cases} 1, & (\delta_x, \delta_y) \in \text{kernel}, \\ 0, & \text{otherwise.} \end{cases}$$

$$\text{Self}-\text{attention:} \quad \boldsymbol{f}'_i = \sum_{j=1}^{N} \alpha_{i,j} \times \boldsymbol{f}_j \cdot \boldsymbol{W}^v, \quad \text{where } a_{i,j} = \frac{\boldsymbol{q}_i \cdot \boldsymbol{k}_j^T}{\sqrt{C'}}, \quad \alpha_{i,j} = \frac{e^{a_{i,j}}}{\sum_{n=1}^{N} e^{a_{i,n}}}.$$

$$\text{Translution:} \quad \boldsymbol{f}'_i = \sum_{j=1}^{N} \alpha_{i,j} \times \boldsymbol{f}_j \cdot \boldsymbol{W}^v_{\delta_x, \delta_y}, \quad \text{where } a_{i,j} = \frac{\boldsymbol{q}_{i,j} \cdot \boldsymbol{k}_{j,i}^T}{\sqrt{C'}}, \quad \alpha_{i,j} = \frac{e^{a_{i,j}}}{\sum_{n=1}^{N} e^{a_{i,n}}}.$$

In other words, convolution and self-attention can be viewed as specific instances of Translution, where convolution simplifies the attention mechanism and self-attention omits the encoding of direction and distance.

## 3.4 LoR-Translution

Suppose there are $H \times W$ input image patches. The relative encoding method in Translution requires $(2H - 1) \times (2W - 1) \times C \times C'$ parameters. Specifically, it requires one parameter matrix $\boldsymbol{W}^q_{\delta_x, \delta_y}$, $\boldsymbol{W}^k_{\delta_x, \delta_y}$ or $\boldsymbol{W}^v_{\delta_x, \delta_y} \in \mathbb{R}^{C \times C'}$ for each relative position $(\delta_x, \delta_y)$, where $\delta_x \in \{-(H-1), \cdots, 0, \cdots, H-1\}$ and $\delta_y \in \{-(W-1), \cdots, 0, \cdots, W-1\}$. This approach leads to excessive parameter demands, making it impractical for most computational devices currently. For instance, in the ViT/16 architecture (Dosovitskiy et al., 2021) with input resolution $224 \times 224$, we have $H = W = \frac{224}{16} = 14$, resulting in $(2H - 1) \times (2W - 1) = 729$ distinct weight matrices for query, key or value. To reduce the number of parameters, inspired by LoRA (Hu et al., 2021), we propose a variant of Translution, *i.e.*, LoR-Translution, which decreases both the input dimension $C$ and the output dimension $C'$ of each $\boldsymbol{W}^q_{\delta_x, \delta_y}$, $\boldsymbol{W}^k_{\delta_x, \delta_y}$, and $\boldsymbol{W}^v_{\delta_x, \delta_y}$, as follows:

$$\boldsymbol{W}^q_{\delta_x, \delta_y} \Rightarrow \boldsymbol{W}^q_1 \cdot \boldsymbol{W}^q_{\delta_x, \delta_y}, \quad \boldsymbol{W}^k_{\delta_x, \delta_y} \Rightarrow \boldsymbol{W}^k_1 \cdot \boldsymbol{W}^k_{\delta_x, \delta_y}, \quad \boldsymbol{W}^v_{\delta_x, \delta_y} \Rightarrow \boldsymbol{W}^v_1 \cdot \boldsymbol{W}^v_{\delta_x, \delta_y} \cdot \boldsymbol{W}^v_2,$$

where $\boldsymbol{W}^q_1 / \boldsymbol{W}^k_1 / \boldsymbol{W}^v_1 \in \mathbb{R}^{C \times C^1}$, $\boldsymbol{W}^q_{\delta_x, \delta_y} / \boldsymbol{W}^k_{\delta_x, \delta_y} / \boldsymbol{W}^v_{\delta_x, \delta_y} \in \mathbb{R}^{C^1 \times C^2}$, $\boldsymbol{W}^v_2 \in \mathbb{R}^{C^2 \times C'}$, and $C^1 \ll C$, $C^2 \ll C'$. Smaller values of $C^1$ and $C^2$ will significantly reduce the number of parameters.

However, setting $C^1$ and $C^2$ too small may overly compress the query, key and value information, negatively impacting performance. To preserve the information, we incorporate the query, key and value computation mechanism of self-attention into LoR-Translution. Specifically, the updated computation is defined as follows:

$$\text{LoR}-\text{Translution} \begin{cases} \text{query encoding:} \quad \boldsymbol{q}_{i,j} = \boldsymbol{f}_i \cdot \boldsymbol{W}^q_1 \cdot \boldsymbol{W}^q_{\delta_x, \delta_y}, \quad \boldsymbol{q}_i = \boldsymbol{f}_i \cdot \boldsymbol{W}^q, \\[2mm] \text{key encoding:} \quad \boldsymbol{k}_{j,i} = \boldsymbol{f}_j \cdot \boldsymbol{W}^k_1 \cdot \boldsymbol{W}^k_{-\delta_x, -\delta_y}, \quad \boldsymbol{k}_j = \boldsymbol{f}_j \cdot \boldsymbol{W}^k, \\[2mm] \text{attention:} \quad a_{i,j} = \dfrac{\boldsymbol{q}_{i,j} \cdot \boldsymbol{k}_{j,i}^T + \boldsymbol{q}_i \cdot \boldsymbol{k}_j^T}{\sqrt{C'}}, \quad \alpha_{i,j} = \dfrac{e^{a_{i,j}}}{\sum_{n=1}^{N} e^{a_{i,n}}}, \\[2mm] \text{value encoding:} \quad \boldsymbol{v}_{i,j} = \boldsymbol{f}_j \cdot (\boldsymbol{W}^v_1 \cdot \boldsymbol{W}^v_{\delta_x, \delta_y} \cdot \boldsymbol{W}^v_2 + \boldsymbol{W}^v), \\[2mm] \text{weighted sum:} \quad \boldsymbol{f}'_i = \sum_{j=1}^{N} \alpha_{i,j} \times \boldsymbol{v}_{i,j}. \end{cases} \quad (2)$$

In this way, LoR-Translution not only possesses relative modeling capability but also reduces the number of parameters.

## 4 EXPERIMENT

In this section, as a fundamental operation, our primary objective is to compare Translution with self-attention, rather than to achieve state-of-the-art performance through specialized network architectures or extensive training techniques. To this end, we conduct experiments using two widely adopted Transformer architectures:

- Vision Transformer (ViT) (Dosovitskiy et al., 2021) for computer vision tasks.
- Generative Pre-trained Transformer (GPT) (Radford et al., 2018; 2019; Brown et al., 2020) for natural language processing tasks. Section C demonstrates how to apply Translution to text modeling.

Table 1 provides an overview of various architecture configures. We substitute self-attention in ViT and GPT with Translution, while maintaining the remaining architecture unchanged.

Table 1: Specifics of architecture configures used in this paper.

| Architecture | Depth (#Layers) | Embedding Dim (Hidden size) | #Heads | MLP Dim (Feedforward) |
|---|---|---|---|---|
| A | 6 | 192 | 3 | 768 |
| B | 12 | 192 | 3 | 768 |
| C | 12 | 384 | 6 | 1,536 |

Due to limited computational resources, our evaluation is primarily conducted on small- and medium-scale architectures. Large-scale evaluation can be performed when single-GPU memory capacities approach approximately $2 \sim 3$ TB. All training starts from scratch. The default compression dimensions for the relative encoding in LoR-Translution are set as $C^1 = C^2 = 8$.

## 4.1 IMAGE CLASSIFICATION WITH VIT

### 4.1.1 DYNAMIC MNIST

To evaluate the capability of modeling relative structure, we synthesize a dynamic MNIST dataset (Srivastava et al., 2015; Fan & Yang, 2019), where digits (originally sized $28 \times 28$ pixels) move within a $84 \times 84$ pixel area, as illustrated in Figure 4. For comparison, we also create a static MNIST dataset of the same size, where digits remain fixed at the center of each image.

Figure 4: Examples of static and dynamic MNIST. Static MNIST digits are fixed at the center of images, whereas dynamic MNIST digits are randomly positioned within the images.

Table 2: Top-1 accuracy (%) on different MNIST settings with the ViT-A architecture. $\mathcal{A} \rightarrow \mathcal{B}$ denotes that models are trained on dataset $\mathcal{A}$ and evaluated on dataset $\mathcal{B}$.

| Arch. | Method | #Params | FLOPs | Static→Static | Dyn→Dyn | Static→Dyn |
|---|---|---|---|---|---|---|
| ViT-A/12 | Self-attention (Vaswani et al., 2017) | 2.7 M | 140.5 M | 98.48 | 92.64 | 18.18 |
| | LoR-Translution (relative dim = 8) | 4.6 M | 146.6 M | 98.48 | 97.31 | 34.90 |
| | Translution | 116.2 M | 140.5 M | **98.60** | **97.35** | **36.40** |
| ViT-A/7 | Self-attention (Vaswani et al., 2017) | 2.7 M | 436.6 M | 98.52 | 93.90 | 19.94 |
| | LoR-Translution (relative dim = 8) | 8.3 M | 454.2 M | 98.81 | 98.57 | 40.05 |
| | Translution | 355.0 M | 436.6 M | **98.91** | **98.60** | **48.07** |

As shown in Table 2, all models achieve high accuracy when trained and evaluated on static MNIST. However, when digit locations vary, the self-attention's accuracy significantly decreases, whereas Translution (including LoR-Translution) still maintains high accuracy. This is because absolute positional embedding makes digit locations part of its representation. Consequently, when digits shift positions, networks may become confused and fail to recognize digits accurately. In contrast, Translution employs relative encoding, effectively capturing digit structures independently of their absolute locations. This significantly reduces sensitivity to location variability, demonstrating Translution's superior capability in modeling relative structures. However, when training on static MNIST, the uniformly black image background causes some $\boldsymbol{W}_{\delta_x,\delta_y}$ not to be well trained. As a result, when evaluated on dynamic MNIST, Translution fails to achieve very high accuracy.

Note that although Translution introduces a larger number of parameters, it does not incur additional computational cost compared with self-attention. As shown in Table 2 and the following tables, Translution uses the same amount of computation as self-attention. This is because Translution simply replaces the original projection matrices $\boldsymbol{W}^q$, $\boldsymbol{W}^k$, and $\boldsymbol{W}^v$ with their offset-indexed counterparts $\boldsymbol{W}^q_{\delta_x,\delta_y}$, $\boldsymbol{W}^k_{\delta_x,\delta_y}$, and $\boldsymbol{W}^v_{\delta_x,\delta_y}$, without increasing the per-token operations. The computational complexity therefore remains unchanged. In contrast, LoR-Translution requires additional computation due to the low-rank adaptation applied during the offset-dependent projection, which increases the processing cost.

Table 3: Accuracy (%) on the ImageNet-1K dataset with patch sizes of 56, 32 and 16. Training is conducted from scratch without pretraining on external datasets, with a batch size of 256.

| Architecture | Method | #Params | FLOPs | Top-1 | Top-5 |
|---|---|---|---|---|---|
| ViT-A/56 | Self-attention (Vaswani et al., 2017) | 4.7 M | 75.86 M | 46.28 | 71.17 |
| | LoR-Translution (relative enc dim = 8) | 5.3 M | 77.92 M | 48.36 | 73.31 |
| | Translution | 38.5 M | 75.86 M | **52.41** | **76.50** |
| ViT-B/56 | Self-attention (Vaswani et al., 2017) | 7.4 M | 121.8 M | 53.75 | 77.59 |
| | LoR-Translution (relative enc dim = 8) | 8.7 M | 126.0 M | 55.87 | 79.16 |
| | Translution | 75.0 M | 121.8 M | **59.51** | **81.97** |
| ViT-C/56 | Self-attention (Vaswani et al., 2017) | 25.3 M | 423.4 M | 64.15 | 84.95 |
| | LoR-Translution (relative enc dim = 8) | 30.5 M | 440.0 M | 66.54 | 86.49 |
| | Translution | 296.0 M | 423.4 M | **68.05** | **88.62** |
| ViT-A/32 | Self-attention (Vaswani et al., 2017) | 3.5 M | 169.0 M | 57.63 | 80.96 |
| | LoR-Translution (relative enc dim = 8) | 5.3 M | 175.0 M | 60.26 | 83.07 |
| | Translution | 116.9 M | 169.0 M | **66.03** | **86.01** |
| ViT-B/32 | Self-attention (Vaswani et al., 2017) | 6.1 M | 308.0 M | 66.13 | 86.87 |
| | LoR-Translution (relative enc dim = 8) | 9.9 M | 320.1 M | 67.63 | 87.96 |
| | Translution | 223.1 M | 308.0 M | **70.63** | **90.10** |
| Translution runs out of memory under the following architectures. | | | | | |
| ViT-C/32 | Self-attention (Vaswani et al., 2017) | 22.9 M | 1.146 G | 73.62 | 91.12 |
| | LoR-Translution (relative enc dim = 8) | 38.0 M | 1.195 G | **74.19** | **91.52** |
| ViT-A/16 | Self-attention (Vaswani et al., 2017) | 3.0 M | 644.8 M | 64.71 | 86.25 |
| | LoR-Translution (relative enc dim = 8) | 10.7 M | 668.6 M | **69.28** | **89.24** |
| ViT-B/16 | Self-attention (Vaswani et al., 2017) | 5.7 M | 1.259 G | 73.51 | 91.89 |
| | LoR-Translution (relative enc dim = 8) | 21.1 M | 1.307 G | **76.20** | **93.04** |
| ViT-C/16 | Self-attention (Vaswani et al., 2017) | 22.0 M | 4.609 G | 78.91 | 94.10 |
| | LoR-Translution (relative enc dim = 8) | 85.4 M | 4.800 G | **79.70** | **94.52** |

### 4.1.2 IMAGENET

ImageNet-1K Deng et al. (2009) is a widely used dataset for computer vision research, particularly in the area of image classification. It contains 1,000 object categories (classes), each with approximately 1,300 training images and 50 validation images, amounting to about 1.28 million training images and 50,000 validation images in total. Images are resized to $224 \times 224$. As shown in Table 3, compared to self-attention (Vaswani et al., 2017), Translution and LoR-Translution effectively improve ImageNet classification.

We compare Translution with existing positional encoding strategies, which typically represent positional information by introducing additional positional biases, as scalars Liu et al. (2021); d'Ascoli et al. (2021) or vectors (Vaswani et al., 2017; Shaw et al., 2018). The formal differences between these approaches are detailed in Section D. As shown in Table 4, compared to existing relative encoding methods, Translution achieves a notable improvement in accuracy.

Table 4: Comparison of different positional encoding strategies. Results are reported on ImageNet-1K with ViT-A/56, trained from scratch (no external pretraining) using a batch size of 256.

| Method | #Parameters | FLOPs | Top-1 | Top-5 |
|---|---|---|---|---|
| Self-attention w/o Pos Emb | 4.69 M | 75.86 M | 42.49 | 67.39 |
| Self-attention w/ Pos Emb (Vaswani et al., 2017) | 4.69 M | 75.86 M | 46.28 | 71.17 |
| Relative key vector (Shaw et al., 2018) | 4.74 M | 75.86 M | 46.39 | 71.25 |
| Relative value vector (Shaw et al., 2018) | 4.74 M | 75.86 M | 46.35 | 71.04 |
| Swin Transformer (Liu et al., 2021) | 4.69 M | 75.86 M | 46.36 | 71.31 |
| ConViT (d'Ascoli et al., 2021) | 4.69 M | 76.04 M | 46.39 | 71.44 |
| RoFormer (Su et al., 2024) | 4.69 M | 75.86 M | 46.65 | 71.51 |
| LoR-Translution | 5.33 M | 77.92 M | 48.36 | 73.31 |
| Translution | 38.53 M | 75.86 M | 52.41 | 76.50 |

### 4.1.3 ABLATION STUDY

*1) Is the improvement of Translution (including LoR-Translution) caused by the introduction of additional parameters or the proposed modeling approach based on relative encoding?*

Compared to self-attention, which employs three parameter matrices $\boldsymbol{W}^q$, $\boldsymbol{W}^k$, $\boldsymbol{W}^v$ to compute query, key and value, Translution uses three groups of parameter matrices $\{\boldsymbol{W}^q_{\delta_x,\delta_y}\}$, $\{\boldsymbol{W}^k_{\delta_x,\delta_y}\}$, $\{\boldsymbol{W}^v_{\delta_x,\delta_y}\}$ for relative encoding, thus introducing more parameters.

To investigate whether the improvement arises from the increased parameter count or from the relative encoding method itself, we conducted the following experiment:

relative encoding: $\boldsymbol{W}^q_{\delta_x,\delta_y}$, $\boldsymbol{W}^k_{\delta_x,\delta_y}$, $\boldsymbol{W}^v_{\delta_x,\delta_y}$ $\Rightarrow$ absolute encoding: $\boldsymbol{W}^q_{i,j}$, $\boldsymbol{W}^k_{i,j}$, $\boldsymbol{W}^v_{i,j}$,

where $\delta_x \in \{-(H-1), \cdots, 0, \cdots, H-1\}$, $\delta_y \in \{-(W-1), \cdots, 0, \cdots, W-1\}$, and indices $i \in [1, H \times W]$ and $y \in [1, H \times W]$. Specifically, for each pair of patches $(i, j)$, a distinct parameter matrix is employed to calculate query, key or value, rather than using the shared offset-based matrices. Under this modification, Translution transitions to absolute modeling. Moreover, this adjustment significantly increases the number of parameter matrices from $(2H-1) \times (2W-1)$ to $(H \times W)^2$.

Table 5: Investigation of whether the improvement of Translution arises from the additional parameters or the proposed relative encoding method ($\boldsymbol{W}^q_{\delta_x,\delta_y}$, $\boldsymbol{W}^k_{\delta_x,\delta_y}$, $\boldsymbol{W}^v_{\delta_x,\delta_y}$). Because the absolute encoding method ($\boldsymbol{W}^q_{i,j}$, $\boldsymbol{W}^k_{i,j}$, $\boldsymbol{W}^v_{i,j}$) consumes a large number of parameters, Translation with ViT-A/7 encounters the out-of-memory issue. Therefore, experiments are conducted using ViT-A/12.

| Method | Encoding | #Parameters | FLOPs | Static→Static | Dynamic→Dynamic | Static→Dynamic |
|--------|----------|-------------|-------|---------------|-----------------|----------------|
| LoR-Translution | relative | 4.6 M | 146.4 M | **98.48** | **97.31** | **34.90** |
| | absolute | 28.7 M | | 98.42 | 96.18 | 25.37 |
| Translution | relative | 116.2 M | 140.5 M | **98.60** | **97.35** | **36.40** |
| | absolute | 1660.9 M | | 98.55 | 53.79 | 11.23 |

As shown in Table 5, although absolute encoding involves significantly more parameters, it achieves lower accuracy than relative encoding. Therefore, simply increasing the number of parameters does not lead to performance improvements.

*2) Impact of relative encoding dimension on the performance of LoR-Translution.*

To reduce parameter usage, LoR-Translution employs smaller input ($C^1$) and output ($C^2$) dimensions for $\{\boldsymbol{W}^q_{\delta_x,\delta_y}\}$, $\{\boldsymbol{W}^k_{\delta_x,\delta_y}\}$ and $\{\boldsymbol{W}^v_{\delta_x,\delta_y}\}$. In our experiments, we set the relative encoding dimensions as $C^1 = C^2 = 8$. This section investigates the impact of varying $C^1$ and $C^2$ on performance. As shown in Table 6, increasing the relative encoding dimension improves accuracy but results in more parameters. Therefore, the relative encoding dimension presents a trade-off between efficiency and effectiveness for LoR-Translution. (When $C^1 = C^2 = 0$, it reduces to self-attention without positional embedding.)

Table 6: Impact of relative encoding dimension on the performance of LoR-Translution with ViT-A/56.

| Rel Enc Dim | #Params | FLOPs | Top-1 | Top-5 | Rel Enc Dim | #Params | FLOPs | Top-1 | Top-5 |
|-------------|---------|-------|-------|-------|-------------|---------|-------|-------|-------|
| $C^1 = C^2 = 0$ | 4.7 M | 75.9 M | 42.49 | 67.39 | $C^1 = C^2 = 8$ | 5.3 M | 77.9 M | 48.36 | 73.31 |
| $C^1 = C^2 = 2$ | 4.7 M | 76.3 M | 46.10 | 71.29 | $C^1 = C^2 = 16$ | 7.0 M | 80.3 M | 48.91 | 73.65 |
| $C^1 = C^2 = 4$ | 4.9 M | 76.8 M | 47.61 | 72.18 | $C^1 = C^2 = 32$ | 13.8 M | 86.2 M | 50.07 | 74.84 |

## 4.2 NATURAL LANGUAGE MODELING WITH GPT

To compare Translution and Transformer for natural language processing, we conduct experiments using the OpenWebText dataset (Gao et al., 2020), an openly available reproduction of OpenAI's proprietary WebText dataset used for GPT-2 (Radford et al., 2019). OpenWebText contains 9 billion training tokens and 4 million validation tokens, with a vocabulary size of 50K. We use perplexity, defined as the exponentiation of the cross-entropy loss, as the evaluation metric, where a lower perplexity indicates stronger language modeling performance. Since the most powerful GPU available to us has 80GB memeory, Translution can handle at most a text sequence of length 160 with the GPT-A architecture. Therefore, we conduct the Translution experiment with sequences of length 160. As shown in Table 7, Translution achieves lower perplexity compared to Transformer, demonstrating its effectiveness in natural language modeling.

We compare Translution with existing positional encoding strategies within the GPT archetecutre on OpenWebText. As shown in Table 8, compared to existing relative encoding methods, Translution achieves a notable decrement in Perplexity.

Table 7: Perplexity on OpenWebText using a batch size of 8 and sequence lengths of 160 and 512.

| Architecture | Method | #Parameters | FLOPs | Perplexity ↓ |
|---|---|---|---|---|
| GPT-A-160 | Self-attention (Vaswani et al., 2017) | 22.0 M | 2.029 G | 60.40 |
| | LoR-Translution (relative enc dim = 8) | 23.7 M | 2.049 G | 57.97 |
| | Translution | 127.5 M | 2.029 G | **56.26** |
| Translution runs out of memory under the following architectures. | | | | |
| GPT-B-160 | Self-attention (Vaswani et al., 2017) | 24.7 M | 2.515 G | 54.82 |
| | LoR-Translution (relative enc dim = 8) | 28.2 M | 2.554 G | **52.72** |
| GPT-C-160 | Self-attention (Vaswani et al., 2017) | 60.0 M | 6.729 G | 39.88 |
| | LoR-Translution (relative enc dim = 8) | 74.0 M | 6.884 G | **39.25** |
| GPT-A-512 | Self-attention (Vaswani et al., 2017) | 22.1 M | 6.910 G | 47.72 |
| | LoR-Translution (relative enc dim = 8) | 27.4 M | 6.972 G | **45.17** |
| GPT-B-512 | Self-attention (Vaswani et al., 2017) | 24.7 M | 8.879 G | 43.18 |
| | LoR-Translution (relative enc dim = 8) | 35.5 M | 9.003 G | **39.92** |

Table 8: Comparison of different positional encoding strategies on OpenWebText with GPT-A-160 and a batch size of 8.

| Method | #Parameters | FLOPs | Perplexity ↓ |
|---|---|---|---|
| Self-attention w/o Pos Emb | 22.0 M | 2.029 G | 74.51 |
| Self-attention w/ Pos Emb (Vaswani et al., 2017) | 22.0 M | 2.029 G | 60.40 |
| Relative key vector (Shaw et al., 2018) | 22.4 M | 2.029 G | 60.13 |
| Relative value vector (Shaw et al., 2018) | 22.4 M | 2.029 G | 59.35 |
| Swin Transformer (Liu et al., 2021) | 22.0 M | 2.029 G | 59.96 |
| ConViT (d'Ascoli et al., 2021) | 22.0 M | 2.032 G | 59.39 |
| RoFormer (Su et al., 2024) | 22.0 M | 2.029 G | 58.02 |
| LoR-Translution | 23.7 M | 2.049 G | 57.97 |
| Translution | 127.5 G | 2.029 G | **56.26** |

## 5 CONCLUSION

In this paper, we introduce Translution, a new operation that unifies self-attention and convolution for adaptive and relative modeling. Experiments on computer vision and natural language processing tasks demonstrate the effectiveness of Translution.

However, due to current limited computational resources, the validation in this paper is preliminary. We encourage the community to further evaluate Translution using larger-scale frameworks and datasets in diverse scenarios to verify its broader applicability, particularly when single GPUs equipped with over $2 \sim 3$ TB of memory are available.

Given Translution's substantial parameter consumption, it is worthwhile to explore optimized variants, such as LoR-Translution. For instance, certain relative positions may share the same parameter, especially when the distance between elements is too long. At the same time, extending Translution to 3D, video, molecule, and other modalities of processing holds significant promise.

As a fundamental operation, Translution can be employed beyond the ViT and GPT architectures. More effective and efficient architectures for Translution merit further exploration in future.

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

## A  DEFAULT NOTATION

| | | | |
|---|---|---|---|
| $a, A$ | A scalar | $\boldsymbol{a}$ | A vector |
| $\boldsymbol{A}$ | A matrix | $\mathbf{A}$ | A tensor |
| $\times$ | Scalar multiplication | $\cdot$ | Matrix multiplication |

## B  GENERAL TRANSLUTION

The calculation of the query, key and value in Translution, *i.e.*, Eq. (1), assumes that element positions (*e.g.*, image patches or textual words) are discrete. In this setting, it is feasible to assign a different set of parameters for each direction and distance. However, if the positions are continuous variables, *e.g.*, in point clouds, it becomes impractical to assign individual weights for each direction and distance, as there are infinitely many possible variations in continuous space. In this case, it may be necessary to design new functions for the relative encoding.

Suppose $\boldsymbol{p}_i$ denotes the position of the $i$-th element. For language, $\boldsymbol{p}_i$ can represent the index of the $i$-th word in the text. For images, $\boldsymbol{p}_i$ corresponds to the row and column indices of the $i$-th patch. For point clouds, $\boldsymbol{p}_i$ refers to the 3D coordinates of the $i$-th point. A more general version of Translution can be formulated as follows,

$$\text{General Translution:}\quad \boldsymbol{f}_i' = \sum_{j=1}^{N} \alpha(\boldsymbol{p}_i - \boldsymbol{p}_j, \boldsymbol{f}_i, \boldsymbol{f}_j,) \times v(\boldsymbol{p}_i - \boldsymbol{p}_j, \boldsymbol{f}_j),$$

where $\alpha \in [0, 1]$ denotes the attention score measuring the relevance of the $j$-th element to the $i$-th element, and $v : \mathbb{R}^{d+C} \to \mathbb{R}^{C'}$ is a function that encodes relative positional information into the element features ($d$ denotes the dimensionality of the position, $C$ is the number of input feature channels, and $C'$ is the number of output feature channels). When applying Translution to a new type of data, the key is to develop effective $\alpha$ and $v$ functions.

## C  1D TRANSLUTION FOR NATURAL LANGUAGE PROCESSING

In the main text, we demonstrate how to apply Translution for image modeling. That Translution can be viewed as a 2D operation because the relative encoding involves two spatial directions. However, in natural language, relative encoding operates along a single dimension, which makes Translution a one-dimensional model when applied to text.

Suppose $\boldsymbol{f}_i \in \mathbb{R}^{1 \times C}$ denotes the embedding (or representation) of the $i$-th token within a text sequence of length $N$, where $C$ represents the embedding dimension. As shown in Figure 5, 1D Translution is designed to integrate adaptive identification of related tokens with relative structural encoding for language modeling. Specifically, Translution retains the self-attention mechanism of the Transformer but employs distinct parameters for computing the Query, Key and Value representations, as follows,

$$\text{1D Translution}\begin{cases}\text{relative query encoding:}\ \boldsymbol{q}_{i,j} = \boldsymbol{f}_i \cdot \boldsymbol{W}_\delta^q,\ \delta = i - j, \\[6pt] \text{relative key encoding:}\ \boldsymbol{k}_{j,i} = \boldsymbol{f}_j \cdot \boldsymbol{W}_{-\delta}^k, \\[6pt] \text{relative attention:}\ a_{i,j} = \dfrac{\boldsymbol{q}_{i,j} \cdot \boldsymbol{k}_{j,i}^T}{\sqrt{C'}},\ \alpha_{i,j} = \dfrac{e^{a_{i,j}}}{\sum_{n=1}^{N} e^{a_{i,n}}}, \\[10pt] \text{relative value encoding:}\ \boldsymbol{v}_{i,j} = \boldsymbol{f}_j \cdot \boldsymbol{W}_\delta^v, \\[6pt] \text{weighted sum:}\ \boldsymbol{f}_i' = \displaystyle\sum_{j=1}^{N} \alpha_{i,j} \times \boldsymbol{v}_{i,j}, \end{cases}$$

where $\boldsymbol{W}_\delta^q / \boldsymbol{W}_\delta^k / \boldsymbol{W}_\delta^v \in \mathbb{R}^{C \times C'}$ denotes the learnable parameters for displacement $\delta$.

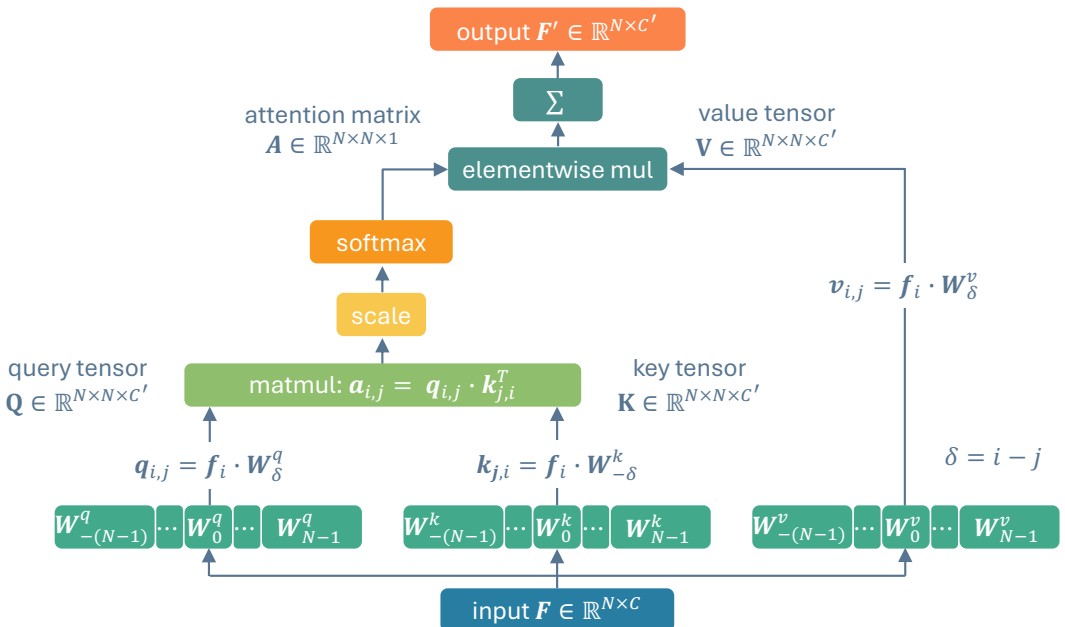

Figure 5: When modeling text, Translution operates in a 1D setting. For a sequence of length $N$, it employs separate parameters for each positional offset (considering both direction and distance), *i.e.*, $\{\boldsymbol{W}^q_{-(N-1)}, \cdots, \boldsymbol{W}^q_0, \cdots, \boldsymbol{W}^q_{N-1}\}$, $\{\boldsymbol{W}^k_{-(N-1)}, \cdots, \boldsymbol{W}^k_0, \cdots, \boldsymbol{W}^k_{N-1}\}$ and $\{\boldsymbol{W}^v_{-(N-1)}, \cdots, \boldsymbol{W}^v_0, \cdots, \boldsymbol{W}^v_{N-1}\}$, to encode relative language structure.

**Causal 1D Translution**

For autoregressive tasks, such as language modeling in GPT, a causal variant is typically required to ensure future tokens remain unseen during inference. In causal 1D Translution, each token attends only to itself and preceding tokens, guaranteeing that predictions rely exclusively on past context, as follows,

$$
\text{Causal 1D Translution}
\begin{cases}
\text{relative query encoding: } \boldsymbol{q}_{i,j} = \boldsymbol{f}_i \cdot \boldsymbol{W}^q_\delta, \ \delta = i - j, \\[4pt]
\text{relative key encoding: } \boldsymbol{k}_{j,i} = \boldsymbol{f}_j \cdot \boldsymbol{W}^k_{-\delta}, \\[4pt]
\text{relative attention: } a_{i,j} = \dfrac{\boldsymbol{q}_{i,j} \cdot \boldsymbol{k}^T_{j,i}}{\sqrt{C'}}, \\[10pt]
\text{causal attention: } a'_{i,j} = \begin{cases} a_{i,j}, & i \geq j, \\ -\infty, & \text{otherwise,} \end{cases} \quad \alpha_{i,j} = \dfrac{e^{a'_{i,j}}}{\sum_{n=1}^N e^{a'_{i,n}}}, \\[14pt]
\text{relative value encoding: } \boldsymbol{v}_{i,j} = \begin{cases} \boldsymbol{f}_j \cdot \boldsymbol{W}_\delta, & \delta = i - j \geq 0, \\ \forall, & \text{otherwise,} \end{cases} \\[14pt]
\text{weighted sum: } \boldsymbol{f}'_i = \sum_{j=1}^N \alpha_{i,j} \times \boldsymbol{v}_{i,j}.
\end{cases}
$$

As shown in Figure 6, compared to the original variant, causal 1D Translution reduces by half the number of parameters needed to compute the query, key and value representations.

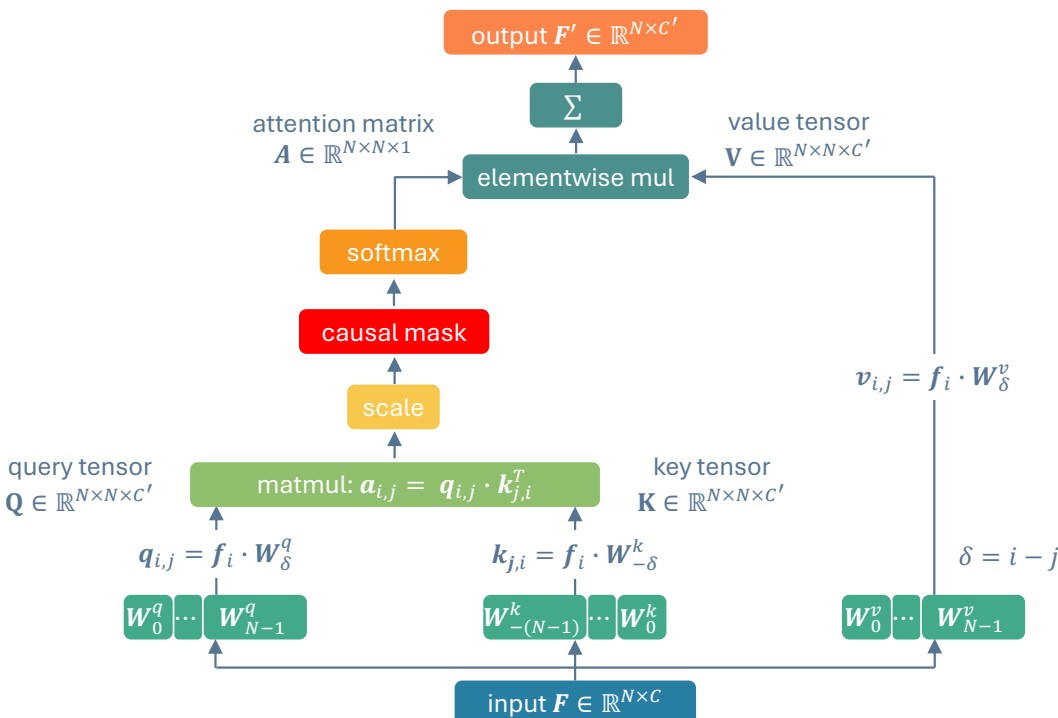

Figure 6: Illustration of causal 1D Translution. For a sequence of length $N$, it employs $N$ parameter matrices to encode relative language structure. Compared to the original 1D Translution, the causal variant reduces the number of parameters required to compute Query, key and Value by half.

## D   MEMORY-EFFICIENT IMPLEMENTATION OF LOR-TRANSLUTION: OPTIMIZING RUNTIME MEMORY USAGE

Recall that $\alpha$-Transformer is defined as follows,

$$\mathrm{LoR-Translution:}\quad \boldsymbol{f}'_i = \sum_{j=1}^{N} \alpha_{i,j} \times \boldsymbol{f}_j \cdot (\boldsymbol{W}^v + \boldsymbol{W}^{v1} \cdot \boldsymbol{W}^v_{\delta_x,\delta_y} \cdot \boldsymbol{W}^{v2}),$$

where $\boldsymbol{W}^v \in \mathbb{R}^{C \times C'}$, $\boldsymbol{W}^{v1} \in \mathbb{R}^{C \times C^1}$, $\boldsymbol{W}^v_{\delta_x,\delta_y} \in \mathbb{R}^{C^1 \times C^2}$, $\boldsymbol{W}^{v2} \in \mathbb{R}^{C^2 \times C'}$, and $C^1 \ll C$, $C^2 \ll C'$. Although this variant significantly reduces the number of parameters, it still demands considerable runtime memory. Specifically, as shown in Figure 3, the resulting value tensor of Translution is $\mathbf{V} \in \mathbb{R}^{N \times N \times C'}$, which is considerably larger than the Transformer's value matrix $\boldsymbol{V} \in \mathbb{R}^{N \times C'}$. To address this issue, we implement LoR-Translution as follows,

$$\boldsymbol{f}'_i = \sum_{j=1}^{N} \alpha_{i,j} \times \boldsymbol{f}_j \cdot \boldsymbol{W}^v + \Big( \sum_{j=1}^{N} \alpha_{i,j} \times \boldsymbol{f}_j \cdot (\boldsymbol{W}^{v1} \cdot \boldsymbol{W}^v_{\delta_x,\delta_y}) \Big) \cdot \boldsymbol{W}^{v2}.$$

This reformulation reduces the peak runtime memory usage from $N \times N \times C'$ to $N \times C' + N \times N \times C^2$, where $C^2 \ll C'$, thus significantly alleviating memory demands during computation.

# E Comparison with Existing Position Modeling Methods

Existing methods typically encode positional information by introducing additional positional biases (either scalars or vectors). In this paper, inspired by convolution, we propose an alternative approach that employs offset-based matrices for relative encoding. In this section, we provide a detailed comparison between these approaches. Suppose $\boldsymbol{x}_i \in \mathbb{R}^{1 \times C}$ represents the feature or representation of the $i$-th patch, located at $(x_i, y_i)$ in an image composed of $N = H \times W$ patches.

*1. Baseline (Self-attention w/o Positional Embedding)*

We consider the self-attention without position embedding as the baseline, formulated as follows:

$$
\begin{aligned}
&\text{w/o input position embedding:} \quad \boldsymbol{f}_i = \boldsymbol{x}_i, \\
&\text{query encoding:} \quad \boldsymbol{q}_i = \boldsymbol{f}_i \cdot \boldsymbol{W}_q, \\
&\text{key encoding:} \quad \boldsymbol{k}_j = \boldsymbol{f}_j \cdot \boldsymbol{W}_k, \\
&\text{attention:} \quad a_{i,j} = \frac{\boldsymbol{q}_i \cdot \boldsymbol{k}_j^T}{\sqrt{C'}}, \quad \alpha_{i,j} = \frac{e^{a_{i,j}}}{\sum_{n=1}^{N} e^{a_{i,n}}}, \\
&\text{value encoding:} \quad \boldsymbol{v}_j = \boldsymbol{f}_j \cdot \boldsymbol{W}_v, \\
&\text{weighted sum:} \quad \boldsymbol{f}_i' = \sum_{j=1}^{N} \alpha_{i,j} \times \boldsymbol{v}_j.
\end{aligned}
$$

*2. Transformer (Self-attention with Positional Embedding)*

Most Transformers, including the original Transformer (Vaswani et al., 2017), employ position embedding to incorporate positional information. Specifically, they integrate absolute positions into element representations, formulated as follows:

$$
\begin{aligned}
&\text{w/ input position embedding:} \quad \boldsymbol{f}_i = \boldsymbol{x}_i + \text{Embed}(x_i, y_i), \\
&\text{query encoding:} \quad \boldsymbol{q}_i = \boldsymbol{f}_i \cdot \boldsymbol{W}_q, \\
&\text{key encoding:} \quad \boldsymbol{k}_j = \boldsymbol{f}_j \cdot \boldsymbol{W}_k, \\
&\text{attention:} \quad a_{i,j} = \frac{\boldsymbol{q}_i \cdot \boldsymbol{k}_j^T}{\sqrt{C'}}, \quad \alpha_{i,j} = \frac{e^{a_{i,j}}}{\sum_{n=1}^{N} e^{a_{i,n}}}, \\
&\text{value encoding:} \quad \boldsymbol{v}_j = \boldsymbol{f}_j \cdot \boldsymbol{W}_v, \\
&\text{weighted sum:} \quad \boldsymbol{f}_i' = \sum_{j=1}^{N} \alpha_{i,j} \times \boldsymbol{v}_j.
\end{aligned}
$$

*3. Relative Key Vector*

Shaw et al. (2018) enhanced Transformer for language modeling by adding learnable relative positional vectors into the key computations. BoTNet (Srinivas et al., 2021) and HaloNet (Vaswani et al., 2021) extended this approach to two dimensions for image processing by adding learnable relative positional vectors into the key computation. This can be formulated as follows,

$$
\begin{aligned}
&\text{w/o input position embedding:} \quad \boldsymbol{f}_i = \boldsymbol{x}_i, \\
&\text{query encoding:} \quad \boldsymbol{q}_i = \boldsymbol{f}_i \cdot \boldsymbol{W}_q, \\
&\text{key encoding:} \quad \boldsymbol{k}_j = \boldsymbol{f}_j \cdot \boldsymbol{W}_k + \boldsymbol{r}_{\delta_x, \delta_y}, \\
&\text{attention:} \quad a_{i,j} = \frac{\boldsymbol{q}_i \cdot \boldsymbol{k}_j^T}{\sqrt{C'}}, \quad \alpha_{i,j} = \frac{e^{a_{i,j}}}{\sum_{n=1}^{N} e^{a_{i,n}}}, \\
&\text{value encoding:} \quad \boldsymbol{v}_j = \boldsymbol{f}_j \cdot \boldsymbol{W}_v, \\
&\text{weighted sum:} \quad \boldsymbol{f}_i' = \sum_{j=1}^{N} \alpha_{i,j} \times \boldsymbol{v}_j,
\end{aligned}
$$

where $\boldsymbol{r}_{\delta_x, \delta_y} \in \mathbb{R}^{1 \times C'}$.

*4. Relative Value Vector*

Shaw et al. (2018) also extended the above relative vector method to the value computations, as follows:

$$
\begin{aligned}
\text{w/o input position embedding:} \ & \boldsymbol{f}_i = \boldsymbol{x}_i, \\
\text{query encoding:} \ & \boldsymbol{q}_i = \boldsymbol{f}_i \cdot \boldsymbol{W}_q, \\
\text{key encoding:} \ & \boldsymbol{k}_j = \boldsymbol{f}_j \cdot \boldsymbol{W}_k, \\
\text{self\ \!-\!attention:} \ & a_{i,j} = \frac{\boldsymbol{q}_i \cdot \boldsymbol{k}_j^T}{\sqrt{C'}}, \ \ \alpha_{i,j} = \frac{e^{a_{i,j}}}{\sum_{n=1}^{N} e^{a_{i,n}}}, \\
\text{value encoding:} \ & \boldsymbol{v}_j = \boldsymbol{f}_j \cdot \boldsymbol{W}_v + \boldsymbol{r}_{\delta_x, \delta_y}, \\
\text{weighted sum:} \ & \boldsymbol{f}_i' = \sum_{j=1}^{N} \alpha_{i,j} \times \boldsymbol{v}_j.
\end{aligned}
$$

*5. Relative Positional Scalar*

Swin Transformer (Liu et al., 2021) and CoAtNet (Dai et al., 2021) incorporate a learnable relative positional bias (a scalar) into the attention score. In these methods, the original self-attention can be regarded as content attention, which measures relationships from the token-feature perspective, while the additional relative positional bias can be regarded as position attention, which measures relationships from the token-position perspective. Formally, this can be expressed as follows:

$$
\begin{aligned}
\text{w/o input position embedding:} \ & \boldsymbol{f}_i = \boldsymbol{x}_i, \\
\text{query encoding:} \ & \boldsymbol{q}_i = \boldsymbol{f}_i \cdot \boldsymbol{W}_q, \\
\text{key encoding:} \ & \boldsymbol{k}_j = \boldsymbol{f}_j \cdot \boldsymbol{W}_k, \\
\text{attention:} \ & a_{i,j} = \frac{\boldsymbol{q}_i \cdot \boldsymbol{k}_j^T}{\sqrt{C'}} + b_{\delta_x, \delta_y}, \ \ \alpha_{i,j} = \frac{e^{a_{i,j}}}{\sum_{n=1}^{N} e^{a_{i,n}}}, \\
\text{value encoding:} \ & \boldsymbol{v}_j = \boldsymbol{f}_j \cdot \boldsymbol{W}_v, \\
\text{weighted sum:} \ & \boldsymbol{f}_i' = \sum_{j=1}^{N} \alpha_{i,j} \times \boldsymbol{v}_j,
\end{aligned}
$$

where $b_{\delta_x, \delta_y} \in \mathbb{R}$. ConViT (d'Ascoli et al., 2021) introduces Gated Positional Self-Attention (GPSA), a variant of self-attention that incorporates a positional inductive bias. Moreover, a learnable gating parameter in each attention head controls the balance between positional and content-based attention, as follows,

$$
\begin{aligned}
\text{w/o input position embedding:} \ & \boldsymbol{f}_i = \boldsymbol{x}_i, \\
\text{query encoding:} \ & \boldsymbol{q}_i = \boldsymbol{f}_i \cdot \boldsymbol{W}_q, \\
\text{key encoding:} \ & \boldsymbol{k}_j = \boldsymbol{f}_j \cdot \boldsymbol{W}_k, \\
\text{patch attention:} \ & a_{i,j} = \frac{\boldsymbol{q}_i \cdot \boldsymbol{k}_j^T}{\sqrt{C'}}, \ \ \alpha_{i,j} = \frac{e^{a_{i,j}}}{\sum_{n=1}^{N} e^{a_{i,n}}}, \\
\text{position attention:} \ & b_{i,j} = \boldsymbol{w} \cdot \boldsymbol{r}_{\|\delta\|}, \ \ \beta_{i,j} = \frac{e^{b_{i,j}}}{\sum_{n=1}^{N} e^{b_{i,n}}}, \\
\text{gated attention:} \ & c_{i,j} = \big(1 - \sigma(\lambda)\big) \times \alpha_{i,j} + \sigma(\lambda) \times \beta_{i,j}, \ \ \xi_{i,j} = \frac{c_{i,j}}{\sum_{n=1}^{N} c_{i,n}}, \\
\text{value encoding:} \ & \boldsymbol{v}_j = \boldsymbol{f}_j \cdot \boldsymbol{W}_v, \\
\text{weighted sum:} \ & \boldsymbol{f}_i' = \sum_{j=1}^{N} \xi_{i,j} \times \boldsymbol{v}_j,
\end{aligned}
$$

where $\boldsymbol{w}$ is a trainable vector for embedding, $\boldsymbol{r}_{\|\delta\|}$ is the relative positional encoding, $\lambda$ is a learnable gate and $\sigma$ is the Sigmoid function.

*6. Rotary Position Embedding*

Unlike the above vector- and scalar-based methods, RoFormer (Su et al., 2024) proposes a rotation-based positional encoding method that is applied directly to queries and keys. As a result, attention scores depend solely on relative distances, eliminating the need to explicitly store a positional vector or scalar, as follows,

$$
\begin{aligned}
&\text{w/o input position embedding:} \ \ \boldsymbol{f}_i = \boldsymbol{x}_i, \\
&\text{query encoding:} \ \ \boldsymbol{q}_i = \boldsymbol{f}_i \cdot \boldsymbol{W}_q, \\
&\text{key encoding:} \ \ \boldsymbol{k}_j = \boldsymbol{f}_j \cdot \boldsymbol{W}_k, \\
&\text{attention:} \ \ \boldsymbol{q}_i', \ \boldsymbol{k}_j' = \text{rotary}(\boldsymbol{q}_i, \ \boldsymbol{k}_j), \ \ a_{i,j} = \frac{\boldsymbol{q}_i' \cdot \boldsymbol{k}_j'^{\,T}}{\sqrt{C'}}, \ \ \alpha_{i,j} = \frac{e^{a_{i,j}}}{\sum_{n=1}^{N} e^{a_{i,n}}}, \\
&\text{value encoding:} \ \ \boldsymbol{v}_j = \boldsymbol{f}_j \cdot \boldsymbol{W}_v, \\
&\text{weighted sum:} \ \ \boldsymbol{f}_i' = \sum_{j=1}^{N} \alpha_{i,j} \times \boldsymbol{v}_j,
\end{aligned}
$$

where $\text{rotary}(\cdot)$ is a rotary position embedding function.

*7. Relative Positional Matrix (Translution)*

Inspired by convolution, we propose Translution that performs matrix multiplication to produce a vector output that encodes displacement or offset information, defined as follows:

$$
\begin{aligned}
&\text{w/o input position embedding:} \ \ \boldsymbol{f}_i = \boldsymbol{x}_i, \\
&\text{relative query encoding:} \ \ \boldsymbol{q}_{i,j} = \boldsymbol{f}_i \cdot \boldsymbol{W}^q_{\delta_x, \delta_y}, \\
&\text{relative key encoding:} \ \ \boldsymbol{k}_{j,i} = \boldsymbol{f}_j \cdot \boldsymbol{W}^k_{-\delta_x, -\delta_y}, \\
&\text{relative attention:} \ \ a_{i,j} = \frac{\boldsymbol{q}_{i,j} \cdot \boldsymbol{k}_{j,i}^{T}}{\sqrt{C'}}, \ \ \alpha_{i,j} = \frac{e^{a_{i,j}}}{\sum_{n=1}^{N} e^{a_{i,n}}}, \\
&\text{relative value encoding:} \ \ \boldsymbol{v}_{i,j} = \boldsymbol{f}_j \cdot \boldsymbol{W}^v_{\delta_x, \delta_y}, \\
&\text{weighted sum:} \ \ \boldsymbol{f}_i' = \sum_{j=1}^{N} \alpha_{i,j} \times \boldsymbol{v}_{i,j}.
\end{aligned}
$$

Table 9 provides a summary of various positional encoding strategies.

Table 9: Summary of different position encoding strategies.

| Method | | |
|---|---|---|
| w/o Pos Emb | $\boldsymbol{f}_i = \boldsymbol{x}_i$ | Baseline |
| w/ Pos Emb | $\boldsymbol{f}_i = \boldsymbol{x}_i + \text{Embed}(x_i, y_i)$ | Transformer (Vaswani et al., 2017) |
| Relative Positional Vector | Key | Shaw et al. (2018), BoTNet (Srinivas et al., 2021), HaloNet (Vaswani et al., 2021), *etc* |
| | Value | Shaw et al. (2018) |
| Relative Positional Scalar | w/o gating | Swin Transformer (Liu et al., 2021), CoAtNet (Dai et al., 2021), *etc* |
| | w/ gating | ConViT (d'Ascoli et al., 2021) |
| Rotary Position Embedding | RoFormer (Su et al., 2024) | |
| Relative Positional Matrix | LoR-Translution | |
| | Translution | |

## F Translution with Input Positional Embedding

In this section, we examine whether incorporating the input positional embedding method from Transformer can further improve Translution. To this end, we implement Translution as follows:

$$
\begin{aligned}
\text{w/ input position embedding:} \quad & \boldsymbol{f}_i = \boldsymbol{x}_i + \text{Embed}(x_i, y_i), \\
\text{relative query encoding:} \quad & \boldsymbol{q}_{i,j} = \boldsymbol{f}_i \cdot \boldsymbol{W}^q_{\delta_x, \delta_y}, \\
\text{relative key encoding:} \quad & \boldsymbol{k}_{j,i} = \boldsymbol{f}_j \cdot \boldsymbol{W}^k_{-\delta_x, -\delta_y}, \\
\text{relative attention:} \quad & a_{i,j} = \frac{\boldsymbol{q}_{i,j} \cdot \boldsymbol{k}^T_{j,i}}{\sqrt{C'}}, \quad \alpha_{i,j} = \frac{e^{a_{i,j}}}{\sum_{n=1}^{N} e^{a_{i,n}}}, \\
\text{relative value encoding:} \quad & \boldsymbol{v}_{i,j} = \boldsymbol{f}_j \cdot \boldsymbol{W}^v_{\delta_x, \delta_y}, \\
\text{weighted sum:} \quad & \boldsymbol{f}'_i = \sum_{j=1}^{N} \alpha_{i,j} \times \boldsymbol{v}_{i,j}.
\end{aligned}
$$

As shown in Table 10, incorporating the Transformer's absolute positional embedding does not yield a clear performance gain for Translution in the static-to-static setting, leads to a slight drop in the dynamic-to-dynamic setting, and results in a substantial drop in the static-to-dynamic setting.

Table 10: Accuracy (%) of Translution w/o and w/ the absolute positional embedding method from Transformer. Results are reported on Static and Dynamic MNIST with ViT-A/12.

| Method | Embed$(x_i, y_i)$ | #Params | FLOPs | Static→Static | Dynamic→Dynamic | Static→Dynamic |
|---|---|---|---|---|---|---|
| LoR-Translution | ✗ | 4.6 M | 146.4 M | 98.48 | **97.31** | **34.90** |
|  | ✓ |  |  | **98.72** | 96.81 | 17.20 |
| Translution | ✗ | 116.2 M | 140.5 M | **98.60** | **97.35** | **36.24** |
|  | ✓ |  |  | 98.47 | 96.31 | 16.50 |

## G Impact of $W^q$, $W^k$ and $W^v$ on LoR-Translution

Recall that: *To reduce the number of parameters, we propose LoR-Translution, which decreases both the input dimension $C^1$ and the output dimension $C^2$ of each $\boldsymbol{W}^q_{\delta_x, \delta_y}$, $\boldsymbol{W}^k_{\delta_x, \delta_y}$, and $\boldsymbol{W}^v_{\delta_x, \delta_y}$. However, setting $C^1$ and $C^2$ too small can overly compress the query, key, and value representations, thereby degrading performance. To address this issue, we integrate the $\boldsymbol{W}^q$, $\boldsymbol{W}^k$, and $\boldsymbol{W}^v$ of Transformer into LoR-Translution to better preserve essential information.*

In this section, we analyze the impact of $\boldsymbol{W}^q$, $\boldsymbol{W}^k$, and $\boldsymbol{W}^v$ by systematically removing them from Eq. (2) as follows:

$$
\text{LoR} - \text{Translution}
\begin{cases}
\text{query encoding:} \quad \boldsymbol{q}_{i,j} = \boldsymbol{f}_i \cdot \boldsymbol{W}^q_1 \cdot \boldsymbol{W}^q_{\delta_x, \delta_y}, \quad \cancel{\boldsymbol{q}_i = \boldsymbol{f}_i \cdot \boldsymbol{W}^q}, \\[4pt]
\text{key encoding:} \quad \boldsymbol{k}_{j,i} = \boldsymbol{f}_j \cdot \boldsymbol{W}^k_1 \cdot \boldsymbol{W}^k_{-\delta_x, -\delta_y}, \quad \cancel{\boldsymbol{k}_j = \boldsymbol{f}_j \cdot \boldsymbol{W}^k}, \\[4pt]
\text{self} - \text{attention:} \quad a_{i,j} = \frac{\boldsymbol{q}_{i,j} \cdot \boldsymbol{k}^T_{j,i} + \cancel{\boldsymbol{q}_i \cdot \boldsymbol{k}^T_j}}{\sqrt{C'}}, \quad \alpha_{i,j} = \frac{e^{a_{i,j}}}{\sum_{n=1}^{N} e^{a_{i,n}}}, \\[4pt]
\text{value encoding:} \quad \boldsymbol{v}_{i,j} = \boldsymbol{f}_j \cdot (\boldsymbol{W}^v_1 \cdot \boldsymbol{W}^v_{\delta_x, \delta_y} \cdot \boldsymbol{W}^v_2 + \cancel{\boldsymbol{W}^v}), \\[4pt]
\text{weighted sum:} \quad \boldsymbol{f}'_i = \sum_{j=1}^{N} \alpha_{i,j} \times \boldsymbol{v}_{i,j}.
\end{cases}
$$

As shown in Table 11, incorporating $\boldsymbol{W}^q$, $\boldsymbol{W}^k$, and $\boldsymbol{W}^v$ significantly enhances the performance of LoR-Translution, particularly when $C^1$ and $C^2$ are small. As $C^1$ and $C^2$ grow larger, the improvement decreases because the information is no longer overly compressed. In this case, $\boldsymbol{W}^q$, $\boldsymbol{W}^k$, and $\boldsymbol{W}^v$ become less critical.

Table 11: Impact of $W^q$, $W^k$ and $W^v$ on LoR-Transformer. Results are reported on ImageNet-1K with ViT-A/56, trained from scratch (no external pretraining) using a batch size of 256.

| Relative Encoding Dimension | $W^q, W^k, W^v$ | #Parameters | FLOPs | Top-1 | Top-5 |
|---|---|---|---|---|---|
| $C^1 = C^2 = 0$ | ✓ | 4.68 M | 75.9 M | 42.49 | 67.39 |
| $C^1 = C^2 = 2$ | ✗ | 4.08 M | 64.4 M | 31.77 | 56.66 |
|  | ✓ | 4.75 M | 76.3 M | 46.10 | 71.29 |
| $C^1 = C^2 = 4$ | ✗ | 4.21 M | 64.9 M | 37.46 | 62.72 |
|  | ✓ | 4.89 M | 76.8 M | 47.61 | 72.18 |
| $C^1 = C^2 = 8$ | ✗ | 4.67 M | 66.0 M | 41.81 | 67.23 |
|  | ✓ | 5.33 M | 77.9 M | 48.36 | 73.31 |
| $C^1 = C^2 = 16$ | ✗ | 6.40 M | 68.4 M | 44.87 | 69.91 |
|  | ✓ | 7.06 M | 80.3 M | 48.91 | 73.65 |
| $C^1 = C^2 = 32$ | ✗ | 13.09 M | 74.3 M | 47.27 | 72.20 |
|  | ✓ | 13.75 M | 86.2 M | 50.07 | 74.84 |

## H  RELATIVE CLS TOKEN

For classification tasks, besides the image tokens, there is an additional CLS token (classification token) that serves as a global representation of the input image. Usually, the CLS token is a learnable embedding appended at the beginning of the input token sequence fed into Transformer. To apply the strategy of relative encoding to the CLS token, we introduce additional parameters: $W^q_{CLS\_in}$, $W^q_{CLS}$, $W^q_{CLS\_out}$, $W^k_{CLS\_in}$, $W^k_{CLS}$, $W^k_{CLS\_out}$, and $W^v_{CLS\_in}$, $W^v_{CLS}$, $W^v_{CLS\_out}$, corresponding to the query, key, and value, respectively.

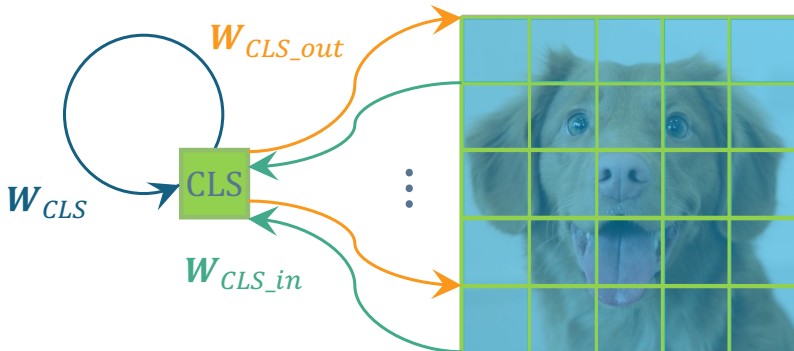

Figure 7: Illustration of relative encoding for the CLS token. For CLS, there are three encoding directions: in, in-place, and out. Correspondingly, three sets of weights, *i.e.*, $W_{CLS\_in}$, $W_{CLS}$, and $W_{CLS\_out}$, are introduced for relative encoding in each respective direction.

As shown in Figure 7, $W_{CLS\_in}$ is utilized when gathering information from the image tokens to update the CLS token; $W_{CLS}$ is applied when updating the CLS token based on its own information; and $W_{CLS\_out}$ is employed when image tokens gather information from the CLS token to update themselves.

