# OpenReview forum: "Translution: Unifying Self-attention and Convolution for Adaptive and Relative Modeling"
_ICLR.cc/2026/Conference — Submitted to ICLR 2026_

### Official Review · Reviewer_5sr9 · 2025-10-25

**Soundness:** 2
**Presentation:** 2
**Contribution:** 2
**Rating:** 2
**Confidence:** 4

**Summary:**

This paper introduces Translution, an operation that unifies self-attention and convolution for adaptive and relative modeling. To mitigate Translution’s substantial parameter overhead, the authors further develop α-Translution. Experiments on computer vision and natural language processing tasks suggest certain effectiveness, though the improvements appear limited. As a general operation, Translution is claimed to be applicable beyond ViT and GPT architectures.

**Strengths:**

This paper proposes Translution, a new operation that unifies self-attention and convolution in a single framework. This idea is conceptually interesting and offers a new perspective on combining adaptive and relative modeling.

The formulation is clearly presented with intuitive figures and mathematical definitions. Experiments across both vision and language tasks show consistent.

The proposed unification maybe inspire future research

**Weaknesses:**

The parameter count of Translution is substantially larger than that of standard self-attention, while the α-Translution structure already includes the self-attention component.  A clearer demonstration of its benefits requires further comparisons under an equal parameter budget.

The experiments have significant limitations. The MNIST dataset is too small and simple. The reported ImageNet accuracy is too low(Table. 4). The validation only used the outdated GPT-2 architecture(Table 7). Because of these issues, the claims about the algorithm's effectiveness are unconvincing.

**Questions:**

N/A

---

> ### Author Response · Authors · 2025-11-23
> **Part I**
>
> Thank you for the constructive comments and for recognizing that our work is **"conceptually interesting and offers a new perspective on combining adaptive and relative modeling"**  and **"the proposed unification maybe inspire future research"**.
>
>
> ## 1. Comparisons under an equal parameter budget.
>
>
> ### (1) **Motivation:** How to further advance artificial neural networks in a future where computational resources are abundant—or even redundant?
>
>
> In recent years, the computational power supporting artificial neural networks has continued to increase. For example, NVIDIA has introduced the B300 GPU, which provides 288 GB of high-bandwidth memory. However, **simply deepening or widening existing networks has brought only limited performance gains.** While computational capacity can, in principle, scale indefinitely, how to enable neural networks to achieve further breakthroughs under abundant computation remains an underexplored question. Without fundamental innovation in network design, it is foreseeable that computation will soon become redundant. Therefore, we aim to explore a key question: **how can we utilize immense computation resources efficiently when they are available, and what new forms of neural networks might emerge from them?**
>
> ### (2) Our goal is not to outperform self-attention under the same parameter or model size.
>
> Based on the above motivation, we would like to emphasize that our goal is not to outperform self-attention under the same parameter or model size. Rather, our objective is to explore how further effectively advance self-attention when abundant computational resources are available. Self-attention has already proven highly effective, and further improving it is inherently challenging. **Therefore, exceeding the performance of self-attention inevitably demands higher computational cost, just as self-attention itself required greater computation to outperform convolution.**
>
> ### (3) Translution can outperform self-attention under an equal parameter budget.
> That being said, we conduct an experiment in which self-attention and LoR-Translution ($\alpha$-Translution) are configured with the same number of parameters. To achieve this, we reduce the channel dimension of LoR-Translution accordingly. As shown in Table 4.1, Translution outperforms self-attention while using the same number of parameters and even fewer FLOPs.
>
> Table 4.1: Accuracy (%) on the ImageNet-1K dataset with the patch size of  56. Training is
> conducted from scratch without pretraining on external datasets, with a batch size of 256.
>
> | Architecture |         Method          |      #Params     |    FLOPs    |   Top-1     |    Top-5     |
> |---------------|-----------------------|------------------|-------------|-------------|-------------|
> | ViT-A/56       |  Self-attention        |       4.7 M         |  75.86 M   | 46.28        | 71.17         |
> | ViT-A/56      | LoR-Translution (relative enc dim = 8) | 4.7 M | 75.16 M     | **46.63** |  **71.50** |

---

> > ### Author Response · Authors · 2025-11-23
> > **Part II**
> >
> > ## 2. More results on ImageNet (Further improve Translution on ImageNet)
> >
> > ### (1) ViT-A/16, ViT-B/16 and ViT-C/16 experiments
> >
> > Table 4.2: Accuracy (%) on the ImageNet-1K dataset with the patch size of 16. Training is conducted from scratch without pretraining on external datasets, with a batch size of 256.
> >
> > | Architecture |         Method          |      #Params     |    FLOPs    |   Top-1     |    Top-5     |
> > |---------------|-----------------------|------------------|-------------|-------------|-------------|
> > | ViT-A/16       |  Self-attention        |       3.0 M         |  644.8 M   | 64.71        | 86.25         |
> > | ViT-A/16      | LoR-Translution (relative enc dim = 8) | 10.7 M | 668.6 M     | **69.28** |  **89.24** |
> > |ViT-B/16      | Self-attention | 5.7 M | 1.259 G | 73.51|  91.89 |
> > |ViT-B/16 |  LoR-Translution (relative enc dim = 8) | 21.1 M | 1.307 G | **76.20** |**93.04**|
> > |ViT-C/16 | Self-attention | 22.0 M | 4.609 G | 78.91 | 94.10 |
> > |ViT-C/16 | LoR-Translution (relative enc dim = 8) | 85.4 M | 4.800 G | **79.70** | **94.52**|
> >
> > ### (2) Integration with Swin-Transformer to further scale up experiment on ImageNet
> >
> > To scale up the experiments, we integrate Translution into the Swin-Transformer architecture. Since Swin-Transformer operates on local windows, it avoids the significant parameter increase typically introduced by full-image attention. Moreover, to further reduce memory consumption, we adopt a variant of LoR-Translution that applies relative encoding only to the value computation.
> >
> > Table 4.3: Accuracy (%) on the ImageNet-1K dataset of integration with Swin-Transformer.
> >
> > | Architecture |         Method          |      #Params     |    FLOPs    |   Acc (%)
> > |---------------|-----------------------|------------------|-------------|-------------|
> > | Swin-T-Base/16       |  Self-attention        |       88 M         |  15.4 M   | 83.5        |
> > | Swin-T-Base/16       | LoR-Translution (relative enc dim = 8) | 92 M | 15.6 M     | **84.1** |
> > | Swin-T-Base/16  + ImageNet-22K Pretrain     |  Self-attention        |       88 M         |  15.4 M   | 85.2        |
> > | Swin-T-Base/16   + ImageNet-22K Pretrain    | LoR-Translution (relative enc dim = 8) | 92 M | 15.6 M     | **85.6** |
> >
> > ### 3. More results on OpenWebText (Longer sequences)
> >
> > GPT-2 supports sequences up to 1,024 tokens, while GPT-3 extends this to 2,048 tokens. In this section, we apply the standard LoR-Translution to 512-token sequences and use a LoR-Translution variant for 4,096-token sequences.
> >
> > ### (1) GPT-A-512 and GPT-B-512 experiments
> >
> > Table 4.4: Perplexity on OpenWebText using a batch size of 8 and a sequence length of 512.
> >
> > | Architecture |         Method          |      #Params     |    FLOPs    |   Perplexity      |
> > |---------------|-----------------------|------------------|-------------|-------------|
> > | GPT-A-512| Self-attention | 22.1 M | 6.910 G | 47.72 |
> > | GPT-A-512| LoR-Translution (relative enc dim = 8) | 27.4 M | 6.972 G | **45.17** |
> > | GPT-B-512| Self-attention | 24.7 M | 8.879 G | 43.18 |
> > | GPT-B-512| LoR-Translution (relative enc dim = 8) | 35.5 M | 9.003 G | **39.92** |
> >
> >
> > ### (2) More Efficient Translution Variant for Scaling Up Experiments on 4096-Token OpenWebText
> >
> > To support longer-context language modeling, we design a lightweight variant of LoR-Translution. First, we apply relative positional encoding only to the value projection, reducing the computational and memory overhead. Second, we restrict the relative-position range to 64 tokens, further improving efficiency while preserving local relative structure.
> >
> > Table 4.5: Perplexity on OpenWebText using a batch size of 8 and a sequence length of 4096.
> >
> > | Architecture |         Method          |      #Params     |    FLOPs    |   Perplexity      |
> > |---------------|-----------------------|------------------|-------------|-------------|
> > | GPT-A-4096| Self-attention | 22.8 M | 89.1 G | 160.77 |
> > | GPT-A-4096| LoR-Translution (relative enc dim = 8) | 33.4 M | 92.2 G | **157.94** |
> > | GPT-B-4096| Self-attention | 25.4 M | 137.8 G | 73.70 |
> > | GPT-B-4096| LoR-Translution (relative enc dim = 8) | 35.5 M | 149.3 G | **68.34** |
> > | GPT-C-4096| Self-attention | 61.5 M | 320.1 G | 49.40 |
> > | GPT-C-4096| LoR-Translution (relative enc dim = 8) | 85.5 M | 340.0 G | **46.44** |
> >
> >
> > Should you need further information, please let us know. We look forward to hearing from you soon. If you feel our efforts address your concerns, we would appreciate it if you could kindly consider raising your rating. Thank you!

---

> ### Comment · Reviewer_5sr9 · 2025-11-25
>
> Thank you very much for the detailed rebuttal. Most of my initial concerns have been addressed. I still believe the paper's overall contribution remains below average.
>
> In the training and application of large models, the number of parameters directly and significantly impacts parallelism and optimization. In the proposed method, the number of model parameters has increased noticeably, yet the accuracy has not improved significantly. It is likely that the accuracy gain is primarily due to the increase in parameters, and a large parameter count presents a significant disadvantage in large-scale models.
>
> A minor issue: The accuracy in Table 4.1 is significantly lower than in Table 4.2. The comparison should be made at a comparable accuracy level.

---

> > ### Author Response · Authors · 2025-11-26
> > **Part III: Thank you!**
> >
> > We thank you for your prompt reply and for raising the rating. We fully respect your decision and your feelings about the contribution. However, we think there may still be a few misunderstandings that we would like to clarify.  We would appreciate the opportunity to gain your further recognition and support.
> >
> > ## 1. Translution uses the same amount of computation as self-attention
> >
> > ## 2. LoR-Translution does not significantly increase the number of parameters
> >
> > ## 3. Translution can outperform self-attention with the same number of parameters and fewer FLOPs.
> > We add a ViT-A/16 experiment so that Table 4.1 and Table 4.2 are presented at comparable levels.
> >
> > Table 4.6: Accuracy (%) on the ImageNet-1K dataset with patch sizes of  56 and 16. To guarantee the same number of parameters, we shorten channels, apply relative encoding only to the value computation and adopt a hybrid design for ViT-A/16–LoR-Translution.
> >
> > | Architecture |         Method          |      #Params     |    FLOPs    |   Top-1     |    Top-5     |
> > |---------------|-----------------------|------------------|-------------|-------------|-------------|
> > | ViT-A/56       |  Self-attention        |       4.7 M         |  75.86 M   | 46.28        | 71.17         |
> > | ViT-A/56      | LoR-Translution (relative enc dim = 8) | 4.7 M | 75.16 M     | **46.63** |  **71.50** |
> > | ViT-A/16       |  Self-attention        |       3.0 M         |  644.8 M | 64.71 |	86.25 |
> > | ViT-A/16      | LoR-Translution (relative enc dim = 8) | 3.0 M	| 642.7 M	| **65.21** | **86.74** |
> >
> > **This also verifies that the improvement does not come from additional parameters; rather, it arises from the relative encoding design in Translution.**
> >
> > ## 4. We should not dismiss a method solely due to its larger parameter count—after all, self-attention itself initially demanded higher computational cost before it could outperform convolution.

---

> > > ### Comment · Reviewer_5sr9 · 2025-11-27
> > >
> > > ImageNet-1k is no longer considered a large-scale dataset. The performance on ImageNet-1k shown in Table 4.6 is insufficient, and this result is not convincing. The review mainly references Table 4.2, where the improvement of LoR-Translution is limited despite a significant increase in the number of parameters. For reference, 'Tokens-to-Token ViT: Training Vision Transformers from Scratch on ImageNet' achieved a Top-1 accuracy of 82.6% four years ago.

---

> > > > ### Author Response · Authors · 2025-11-27
> > > > **Part IV: It’s a pity that you raised the rating but later reverted it.**
> > > >
> > > > Thank you for the feedback. We feel it is a pity that the rating was raised but later reverted, though we fully respect your decision. However, as we noted in our responses to other comments, the Translution variants can in fact achieve higher accuracy.
> > > >
> > > > ## 1. Integration with Swin-Transformer to further scale up experiment on ImageNet
> > > >
> > > > Table 4.7: Accuracy (%) on the ImageNet-1K dataset of integration with Swin-Transformer.
> > > > | Architecture |         Method          |      #Params     |    FLOPs    |   Acc (%)
> > > > |---------------|-----------------------|------------------|-------------|-------------|
> > > > | Swin-T-Base/16       |  Self-attention        |       88 M         |  15.4 M   | 83.5        |
> > > > | Swin-T-Base/16       | LoR-Translution (relative enc dim = 8) | 92 M | 15.6 M     | **84.1** |
> > > > | Swin-T-Base/16  + ImageNet-22K Pretrain     |  Self-attention        |       88 M         |  15.4 M   | 85.2        |
> > > > | Swin-T-Base/16   + ImageNet-22K Pretrain    | LoR-Translution (relative enc dim = 8) | 92 M | 15.6 M     | **85.6** |
> > > >
> > > > We hope our response addresses your concerns.

---

### Official Review · Reviewer_LDCV · 2025-10-29

**Soundness:** 3
**Presentation:** 3
**Contribution:** 2
**Rating:** 4
**Confidence:** 4

**Summary:**

This paper introduces Translution, a new neural operation intended to unify the adaptive region selection of self-attention with the relative structural encoding of convolution.
The authors argue that self-attention excels at adaptive relevance modeling but depends on absolute positional embeddings, while convolution naturally encodes relative spatial structure but has a fixed receptive field.
Translution extends self-attention by assigning separate learnable matrices for each spatial offset ($\delta x, \delta y$) in query, key, and value computations, thereby achieving relative encoding.
A lightweight variant, α-Translution, is proposed to reduce parameter count.
Experiments on vision (ViT) and language (GPT) architectures show consistent accuracy gains over standard self-attention, especially in tasks involving positional shifts (e.g., dynamic MNIST).

**Strengths:**

1. Overall clarity:  The paper is very clear and fluent, it was a pleasure to read it.
2. Conceptual clarity and motivation: The paper clearly articulates the complementary strengths of convolution (relative encoding) and self-attention (adaptive selection) and unifies them in a principled way.
3. Novel formulation: Translution generalizes both convolution and self-attention as special cases, offering an elegant theoretical unification that may inspire new architectural directions.
4. Empirical evidence for relative modeling: The dynamic MNIST and ImageNet experiments convincingly demonstrate that Translution is more invariant to translation and better at modeling relative structure.
5. Comprehensive comparisons: The paper includes baselines using multiple positional encoding schemes (absolute, relative vector/scalar, RoPE), showing consistent improvements.

**Weaknesses:**

1. Impracticality for LLMs: the Translution method with model weights which are proportional to max_sequence_length is impractical for LLMs with sequence lengths $N$ reaching  ~10M tokens. Both in model parameters and runtime. Even when considering $\alpha$-Translution with small $C^1$ and $C^2$, the model weights still have a dependency on $N$

2. What about runtime performance? how will Translution/$\alpha$-translution affect the latency of inference. Specifically, as a function of sequence length. This measurement has a tremendous factor on acceptance of the method.

3. Computational cost and scalability: Translution by-itself introduces a very large number of parameters, making it impractical for modern-scale models. I will consider it only as the hypothetical performance limit of this methodology. Nevertheless, part of this high limit stems from the mere addition of model parameters.
the ablation study of section 4.1.3 does not fully convince me about this issue. The fact that a 1B parameter model did not reach a high accuracy may stem from the lack of training data or amount of cycles.

4. Limited experimental scale: Most results are on small or medium-sized ViT/GPT architectures. Especially GPT with the very short sequence length (160…) The authors acknowledge memory constraints but this limits confidence in real-world applicability for Small Language Models

5. Reference to the Positional embedding as described by Vaswani et al. all along the work:
Most of the discussion and reference is to an obsolete Positional embedding standard. In the last few years, RoPE has become the de-facto positional embedding which is reapplied in each layer.
Also, as reported in “Rotary Position Embedding for Vision Transformer”, for high resolution images, RoPE improves accuracy without introduction of additional model parameters…

**Questions:**

$Q_1$ The reduction of dimensionality to $C^1$ and $C^2$ reminds me of LoRA. Is it simply it? Please refer to it.

$Q_2$ How does $\alpha$-Translution’s training and inference runtime scale compared to standard relative self-attention for large-scale ViTs or GPTs?

$Q_3$ Can the proposed operation be efficiently implemented via low-rank or factorized parameter sharing (e.g., learned basis over offsets)?

$Q_4$ The layers are not identical in their representation abilities. The first layers are more sensitive to modifications while the middle layers are more compressible. In this sense, would a hybrid design (Translution in early layers, vanilla-self-attention in later ones) retain most benefits while reducing cost?

---

> ### Author Response · Authors · 2025-11-23
> **Part I**
>
> Thank you for the constructive comments and for recognizing the **"novel formulation"**, **"comprehensive comparisons"**, and the **"conceptual clarity and motivation"** of our work.
>
> ## 1. Handle long sequence
>
> As mentioned in the conclusion, we can design more efficient and lightweight variants to better handle long sequences: **”For instance, certain relative positions may share the same parameter, especially when the distance between elements is too long.”**
> Here, to support longer-context language modeling, we additionally design a lightweight LoR-Translution variant. First, we apply relative positional encoding only to the value projection, significantly reducing computation and memory usage. Second, we limit the relative-position window to 64 tokens, which further improves efficiency while preserving essential local relational structure.
>
> Table 3.1: Perplexity on OpenWebText using a batch size of 8 and a sequence length of 4096.
>
> | Architecture |         Method          |      #Params     |    FLOPs    |   Perplexity      |
> |---------------|-----------------------|------------------|-------------|-------------|
> | GPT-A-4096| Self-attention | 22.8 M | 89.1 G | 160.77 |
> | GPT-A-4096| LoR-Translution (relative enc dim = 8) | 33.4 M | 92.2 G | **157.94** |
> | GPT-B-4096| Self-attention | 25.4 M | 137.8 G | 73.70 |
> | GPT-B-4096| LoR-Translution (relative enc dim = 8) | 35.5 M | 149.3 G | **68.34** |
> | GPT-C-4096| Self-attention | 61.5 M | 320.1 G | 49.40 |
> | GPT-C-4096| LoR-Translution (relative enc dim = 8) | 85.5 M | 340.0 G | **46.44** |
>
> We believe that as GPUs continue to improve, more powerful Translution variants will become feasible and can be applied to handle much longer sequences.
>
> ## 2. Runtime performance: **Translution uses the same amount of computation as self-attention**
>
> Although Translution introduces a larger number of parameters, it does not incur additional computational cost compared with  self-attention. As shown in tables, Translution uses the same amount of computation as self-attention. This is because Translution simply replaces the original projection matrices $\boldsymbol{W}^q$, $\boldsymbol{W}^k$, and $\boldsymbol{W}^v$ with their offset-indexed counterparts $\boldsymbol{W}^q_{\delta_x,\delta_y}$, $\boldsymbol{W}^k_{\delta_x,\delta_y}$, and $\boldsymbol{W}^v_{\delta_x,\delta_y}$, without increasing the per-token operations.
> The computational complexity therefore remains unchanged.
> In contrast, LoR-Translution requires additional computation due to the low-rank adaptation applied during the offset-dependent projection, which increases the processing cost.
>
> However, Translution still incurs longer inference time because it requires additional GPU memory access or indexing of the offset-dependent weights $\boldsymbol{W}_{\delta_x,\delta_y}$.
>
> Table 3.2: Inference time with 224×224 input images. Inference time = Computation time +  GPU memory access time
>
> | Architecture |         Method          |      #Params     |    FLOPs    |   Computation time     |   GPU memory access time / $\boldsymbol{W}_{\delta_x,\delta_y}$ indexing time |
> |---------------|-----------------------|------------------|-------------|-------------|-------------|
> | ViT-A/56       |  Self-attention        |       4.7 M |75.86 M|      1.2 ms   |   0.1 ms  |
> | ViT-A/56      | LoR-Translution | 5.3 M | 77.92 M | 1.3 ms | 0.6 ms |
> | ViT-A/56      | Translution | 38.5 M | 75.86 M |  1.2 ms | 1.1 ms |

---

> > ### Author Response · Authors · 2025-11-23
> > **Part II**
> >
> > ## 3. Scalability
> > (1)	Vision. The input image size or resolution is naturally limited. Current GPUs can already support Translution on 224×224 images—the widely used resolution in vision benchmarks. Therefore, we conduct comprehensive experiments on image tasks and show that both Translution and LoR-Translution are effective for visual processing.
> >
> > (2)	Language. Current GPUs do make it challenging for Translution to handle very long sequences. However, as discussed above, we can mitigate this by sharing parameters for tokens that are sufficiently far apart, reducing both memory usage and computational cost.
> >
> > (3)	GPUs will continue to improve over time. For example, NVIDIA has introduced the B300 GPU, which provides 288 GB of high-bandwidth memory. This can be more and more improved sooner in the future. In fact, our work aims to explore a broader question: **How can we further advance artificial neural networks in a future where computational resources are abundant—or even no longer a limiting factor? Beyond simply stacking deeper and deeper Transformers, we investigate whether alternative architectural directions—such as Translution—can further push the capabilities of deep neural networks.**
> >
> > (4)	About absolute Translution (Section 4.1.3)
> >
> > We train the absolute Translution variant using a larger-scale dataset (1.2 million images) and significantly more training epochs (increase to 600 and 900). All experiments in this section are conducted with the ViT-C/56 architecture. As shown in Table 3.3, larger-scale dataset and more training epochs do not significanly improve absolute Translution.
> >
> > Table 3.3: Accuracy (%) on the ImageNet-1K dataset with a patch size of 56. Training is conducted from scratch without pretraining on external datasets, with a batch size of 256.
> > | Method | Encoding | #Params     |    FLOPs    |   Top-1     |    Top-5     |
> > |---------------|-----------|----------------|-------------|-------------|-------------|
> > |LoR-Translution| relative (300 epochs) | 30.5 M | 440.0 M | **66.54** | **86.49** |
> > |LoR-Translution| absolute (300 epochs) | 50.1 M | 440.0 M | 42.252 | 67.282 |
> > |LoR-Translution| absolute (600 epochs) | 50.1 M | 440.0 M | 42.314 | 67.598 |
> > |LoR-Translution| absolute (900 epochs) | 50.1 M | 77.92 M | 42.376 | 67.618 |
> > |Translution| relative (300 epoches) | 296.0 M | 423.4 M | **68.05** | **88.62** |
> > |Translution| absolute (300 epoches) | 1554.1 M | 423.4  M | 38.666 | 63.888 |
> > |Translution| absolute (600 epoches) | 1554.1 M | 423.4  M | 40.280 |65.452 |
> > |Translution| absolute (900 epoches) | 1554.1 M | 423.4  M | 41.292 | 66.294 |
> >
> > ## 4.  Scaling up experiments
> >
> > Besides the above GPT-4096 experiments, we also conduct additional scaled-up ViT experiments.
> >
> > ### (1) ViT-A/16, ViT-B/16 and ViT-C/16 experiments
> >
> > Table 3.3: Accuracy (%) on the ImageNet-1K dataset with the patch size of  16. Training is
> > conducted from scratch without pretraining on external datasets, with a batch size of 256.
> >
> > | Architecture |         Method          |      #Params     |    FLOPs    |   Top-1     |    Top-5     |
> > |---------------|-----------------------|------------------|-------------|-------------|-------------|
> > | ViT-A/16       |  Self-attention        |       3.0 M         |  644.8 M   | 64.71        | 86.25         |
> > | ViT-A/16      | LoR-Translution (relative enc dim = 8) | 10.7 M | 668.6 M     | **69.28** |  **89.24** |
> > |ViT-B/16      | Self-attention | 5.7 M | 1.259 G | 73.51|  91.89 |
> > |ViT-B/16 |  LoR-Translution (relative enc dim = 8) | 21.1 M | 1.307 G | **76.20** |**93.04**|
> > |ViT-C/16 | Self-attention | 22.0 M | 4.609 G | 78.91 | 94.10 |
> > |ViT-C/16 | LoR-Translution (relative enc dim = 8) | 85.4 M | 4.800 G | **79.70** | **94.52**|
> >
> >
> > ### (2)  Integration with Swin-Transformer to further scale up experiment on ImageNet
> >
> > To scale up the experiments, we integrate Translution into the Swin-Transformer architecture. Since Swin-Transformer operates on local windows, it avoids the significant parameter increase typically introduced by full-image attention. Moreover, to further reduce memory consumption, we adopt a variant of LoR-Translution that applies relative encoding only to the value computation.
> >
> > Table 3.4: Accuracy (%) on the ImageNet-1K dataset of integration with Swin-Transformer.
> > | Architecture |         Method          |      #Params     |    FLOPs    |   Acc (%)
> > |---------------|-----------------------|------------------|-------------|-------------|
> > | Swin-T-Base/16       |  Self-attention        |       88 M         |  15.4 M   | 83.5        |
> > | Swin-T-Base/16       | LoR-Translution (relative enc dim = 8) | 92 M | 15.6 M     | **84.1** |
> > | Swin-T-Base/16  + ImageNet-22K Pretrain     |  Self-attention        |       88 M         |  15.4 M   | 85.2        |
> > | Swin-T-Base/16   + ImageNet-22K Pretrain    | LoR-Translution (relative enc dim = 8) | 92 M | 15.6 M     | **85.6** |

---

> ### Author Response · Authors · 2025-11-23
> **Part III**
>
> ## 5.  Comparison with  relative positional encodings, e.g. RoPE, on natural language modeling.
>
> We compare Translution with existing positional encoding strategies within the GPT archetecutre on OpenWebText. As shown in Table 3.5, compared to existing relative encoding methods, Translution achieves a notable decrement in Perplexity. These. Results are also included in the updated version (Table 10).
>
> Table 3.5: Comparison of different positional encoding strategies on OpenWebText with GPT-A-160 and a batch size of 8.
> Method | #Parameters | FLOPs | Perplexity ↓|
> |---------------|-----------------------|------------------|-------------|
> |Self-attention (w/o Pos Emb) | 22.0 M | 2.029 G | 74.51 |
> |Self-attention  (w/ Pos Emb) (Vaswani et al., 2017) | 22.0 M | 2.029 G | 60.40 |
> | Relative key vector (Shaw et al., 2018) | 22.4 M | 2.029 G | 60.13 |
> | Relative value vector (Shaw et al., 2018) | 22.4 M | 2.029 G | 59.35 |
> | Swin Transformer (Liu et al., 2021) | 22.0 M | 2.029 G | 59.96 |
> | ConViT (d’Ascoli et al., 2021) | 22.0 M | 2.032 G | 59.39 |
> | RoFormer / RoPE (Su et al., 2024) | 22.0 M | 2.029 G |  58.02 |
> | LoR-Translution | 23.7 M | 2.049 G | 57.97 |
> | Translution | 127.5 G | 2.029 G | 56.26 |
>
> ## 6. Questions
>
> ### (1) LoRA
>
> Thanks for the suggestion. α-Translution shares a similar intuition with LoRA.
> LoRA’s primary purpose is, without altering the original model weights, to enable fine-tuning of large pretrained models.
> In contrast, our goal is to reduce parameter count, and we do not require the model weights to remain unchanged.
> **Given the similarity in form, we rename α-Translution to LoR-Translution to make its design and motivation clearer to readers.**
>
> ### (2) How does α-Translution’s training and inference runtime scale compared to standard relative self-attention for large-scale ViTs or GPTs?
>
> First, we would like to emphasize that there is no such thing as a "standard relative self-attention".
> Second, the training and inference runtimes of self-attention and RoPE are reported in Table 3.6.
>
> Table 3.6: Training and inference runtime comparison.
> | Architecture |  Method  |     Training  time | Inference time |
> |---------------|-----------------|---------------------|---------------------|
> | ViT-A/56    |  Self-attention   |   3.6 ms | 1.3 ms |
> | ViT-A/56    |  RoFormer (RoPE)   |   4.1 ms | 1.5 ms |
> | ViT-A/56    | LoR-Translution  |   4.9 ms | 1.9 ms |
> | GPT-A-512    |  Self-attention   |   5.0 ms | 1.7 ms |
> | GPT-A-512    |  RoFormer (RoPE)   |   7.2 ms | 2.4 ms |
> | GPT-A-512    | LoR-Translution  |   7.9 ms | 2.7 ms |
>
> ### (3) Learned basis over offsets.
>
> These methods can be seen as Relative key vector (Shaw et al., 2018), Relative value vector (Shaw et al., 2018) , Swin Transformer (Liu et al., 2021) and ConViT (d’Ascoli et al., 2021). As shown in Table 4 in the main paper, they do not effectively improve performance.
>
> ### (4) Hybrid design.
>
> Thank you for the great suggestion. As shown in Table 3.7, even a single Translution layer already outperforms standard self-attention (with positional embeddings). However, for the ViT-A/56 architecture, applying Translution to the middle layers appears to yield greater benefits. We will further explore this hybrid design in future work.
>
> Table. 3.7: Accuracy (%) on the ImageNet-1K dataset with Hybrid  ViT-A/56 (Translution + Self-attention). Training is
> conducted from scratch without pretraining on external datasets, with a batch size of 256.
> | Architecture | Top-1 | Top-5 |
> | ------------|---------|---------|
> | Self-attention| 57.63 | 80.96 |
> | 1st Translution |57.25|80.55|
> | 2nd Translution |59.76|82.71|
> | 3rd Translution |60.07|82.82|
> | 4th Translution |60.14 | 82.89|
> | 5th Translution |59.76 | 82.53|
>
> Should you need further information, please let us know. We look forward to hearing from you soon. If you feel our efforts address your concerns, we would appreciate it if you could kindly consider raising your rating. Thank you!

---

> > ### Comment · Reviewer_LDCV · 2025-11-24
> > **Reply to rebuttal**
> >
> > I thank the authors for the rebuttal. most of my concerns have been approached.
> > I originally scored the work as good for both sound and presentation.
> > It is well written and clear to follow. It is also innovative in its approach.
> > Nevertheless, I still think its contribution is below average, because the results do not exceed standard RoPE mechanism.
> > Let’s divide the general scope to two main sub-domains: a. vision with relatively small sequence length, and b. NLP with very long sequence length N>=10^6 tokens.
> >
> > For the first sub domain -- Vision, it is possible to outperform standard attention even with Linear attention – i.e., reducing compute, with results that exceeds the results of standard attention as well as that of Translution
> > For NLP, it is unacceptable to maintain model weights which are of order ~N. In addition, it is not shown that this method outperforms RoPE. (see “Comparison with relative positional encodings, e.g. RoPE, on natural language modeling.”)
> > Also in section 4. Of the rebuttal:
> > The table indeed shows a small improvement in the accuracy – but it’s one that would be expected by any 4x increase of the model parameters (and not necessarily translation)

---

> ### Author Response · Authors · 2025-11-25
> **Part IV**
>
> We thank you for  your prompt reply !  However, we think there may be some misunderstanding here.
>
> ## 1. The statement "It is not shown that this method outperforms RoPE" appears to be a misunderstanding. In fact, our method **does outperform** RoPE for both the ViT and GPT settings, as demonstrated in Table 4 of the original submission, as well as in Table 3.5 of our responses.
>
> ## 2. In section 4. Of the rebuttal (Scaling up experiments): The table indeed shows a small improvement in the accuracy.
>
> Improving accuracy on the large-scale ImageNet dataset is extremely challenging. For example, as reported in the ICCV Best Paper Swin Transformer, the state-of-the-art accuracy only increased from 82.9% (RegNetY-16G) to 83.5% (Swin Transformer) under the 224×224 resolution. In contrast, even our simplified variant of LoR-Translution surpasses Swin Transformer, improving accuracy from 83.5% to 84.1%.
>
> ## 3. Translution variant can handle very long sequences.
>
> As shown in Section 1 of our responses, by limiting the relative-position window size, we can effectively handle long sequences.  In this way, the number of parameters does not increase as the sequence length grows. This strategy has also been empirically validated in experiments.
>
> We hope our response addresses your concerns.

---

### Official Review · Reviewer_kArY · 2025-10-31

**Soundness:** 3
**Presentation:** 3
**Contribution:** 3
**Rating:** 6
**Confidence:** 4

**Summary:**

The paper proposes Translution, a novel advanced attention mechanism that unifies the adaptive nature of self-attention with the relative encoding ability of convolution. A lightweight version, $/alpha$-Translution, reduces parameter cost. Experiments on ViT and GPT architectures demonstrate improved accuracy on both CV and NLP tasks.

**Strengths:**

* The motivation is clear and the method is novel.

* The writing is clear.

* The derivation of Translution is elegant and connects convolution and self-attention through a unified mathematical framework.

* Strong empirical results across both vision and language domains, with meaningful ablation analyses.

**Weaknesses:**

* The experiment on LM lacks baselines of relative positional encodings, e.g. RoPE.

* The proposed parameterization will siginificantly increase the number of parameters when the input sequence length is big.

**Questions:**

Pls refer to Weaknesses. I'd like to increase my scores if the additional experiments could be updated.

---

> ### Author Response · Authors · 2025-11-23
> **Thank you!**
>
> Thank you for the constructive comments and for acknowledging that **"the method is novel"**, **"the derivation of Translution is elegant"**, and **"strong empirical results"**.
>
> ## 1. Experiment on LM baselines with relative positional encodings, e.g. RoPE.
>
> We compare Translution with existing positional encoding strategies within the GPT architecture on OpenWebText. As shown in Table2.1, compared to existing relative encoding methods, Translution achieves a notable decrement in Perplexity. These. Results are also invluded in the updated version (Table 10).
>
> Table 2.1: Comparison of different positional encoding strategies on OpenWebText with GPT-A-160 and a batch size of 8.
> Method | #Parameters | FLOPs | Perplexity ↓|
> |---------------|-----------------------|------------------|-------------|
> |Self-attention (w/o Pos Emb ) | 22.0 M | 2.029 G | 74.51 |
> |Self-attention  (w/ Pos Emb) (Vaswani et al., 2017) | 22.0 M | 2.029 G | 60.40 |
> | Relative key vector (Shaw et al., 2018) | 22.4 M | 2.029 G | 60.13 |
> | Relative value vector (Shaw et al., 2018) | 22.4 M | 2.029 G | 59.35 |
> | Swin Transformer (Liu et al., 2021) | 22.0 M | 2.029 G | 59.96 |
> | ConViT (d’Ascoli et al., 2021) | 22.0 M | 2.032 G | 59.39 |
> | RoFormer / RoPE (Su et al., 2024) | 22.0 M | 2.029 G |  58.02 |
> | LoR-Translution | 23.7 M | 2.049 G | 57.97 |
> | Translution | 127.5 G | 2.029 G | 56.26 |
>
> ##2. When the input sequence length is big.
>
> As noted in the conclusion, applying Translution to very large inputs currently requires substantial GPU memory. This limitation may be gradually alleviated as increasingly powerful hardware becomes available. For example, NVIDIA’s latest B300 GPU already provides nearly 300 GB of HBM. On the algorithmic side, developing more efficient and lightweight variants of Translution is a promising direction for enabling large-context processing, though this is beyond the primary scope of the present work.
>
> To support longer-context language modeling, we additionally design a lightweight LoR-Translution variant. First, we apply relative positional encoding only to the value projection, significantly reducing computation and memory usage. Second, we limit the relative-position window to 64 tokens, which further improves efficiency while preserving essential local relational structure.
>
> Table 2.2: Perplexity on OpenWebText using a batch size of 8 and a sequence length of 4096.
> | Architecture |         Method          |      #Params     |    FLOPs    |   Perplexity      |
> |---------------|-----------------------|------------------|-------------|-------------|
> | GPT-A-4096| Self-attention | 22.8 M | 89.1 G | 160.77 |
> | GPT-A-4096| LoR-Translution (relative enc dim = 8) | 33.4 M | 92.2 G | **157.94** |
> | GPT-B-4096| Self-attention | 25.4 M | 137.8 G | 73.70 |
> | GPT-B-4096| LoR-Translution (relative enc dim = 8) | 35.5 M | 149.3 G | **68.34** |
> | GPT-C-4096| Self-attention | 61.5 M | 320.1 G | 49.40 |
> | GPT-C-4096| LoR-Translution (relative enc dim = 8) | 85.5 M | 340.0 G | **46.44** |
>
> Should you need further information, please let us know. We look forward to hearing from you soon. If you feel our efforts address your concerns, we would appreciate it if you could kindly consider raising your rating. Thank you!

---

> > ### Comment · Reviewer_kArY · 2025-11-24
> > **Thanks for the additional experiments**
> >
> > The updated table has shown strong results and addresses my concern succesfully. I will increase my score accordingly.

---

> > > ### Author Response · Authors · 2025-11-25
> > > **Thank you again!**
> > >
> > > We sincerely appreciate your prompt reply and your recognition of our efforts. Thank you again!

---

### Official Review · Reviewer_uvaq · 2025-11-01

**Soundness:** 3
**Presentation:** 3
**Contribution:** 2
**Rating:** 6
**Confidence:** 5

**Summary:**

This paper proposes an improved self-attention operator named "translution," which essentially replaces the Q, K, and V in the vanilla self-attention mechanism with a location-dependent convolutional operator. However, directly applying this paradigm would lead to an explosion in the number of parameters. To address this, the authors decompose the location-dependent convolutional operator in a manner similar to depthwise separable convolution, thereby significantly reducing the parameter count. Experimental results demonstrate that this operator achieves moderate yet consistent improvements over the vanilla self-attention mechanism.

**Strengths:**

The greatest strength of this paper lies in the clear effectiveness demonstrated by the experimental results of the proposed operator. It provides a relatively comprehensive set of experimental results. The paper as a whole is clearly written and well-structured, making it easy to follow.

**Weaknesses:**

The primary issue with this paper is overclaiming. The authors suggest that this operator has the potential to replace self-attention. However, based on the elaboration of the core idea, it essentially adds a location-dependent convolutional operator to self-attention. Methods employing similar techniques are already quite prevalent; for example: [1][2][3], where [3] replaces the K, Q, V in self-attention with K*K convolutions.

As we know, a crucial property of the Transformer is its scaling capability. I am very curious about whether Translution maintains a performance advantage when scaled up, and whether the added convolution remains genuinely important in large-scale models. Therefore, the authors should add scaling experiments to demonstrate this characteristic.

Furthermore, the training configuration has a critical impact on final performance. Were all experiments in this paper conducted under the same training settings? From the experiments, the model scale for Translution seems significantly larger than the baseline models. To what extent does this difference in scale contribute to the performance improvement? This point needs clarification through experimental analysis.

Additionally, incorporating a location-dependent convolution might lead to a significant decrease in inference efficiency. The authors should provide comparative data on inference efficiency.

[1] CoAtNet: Marrying Convolution and Attention for All Data Sizes

[2] X-volution: On the unification of convolution and self-attention

[3] Restormer: Efficient Transformer for High-Resolution Image Restoration

**Questions:**

Please refer to Weaknesses

---

> ### Author Response · Authors · 2025-11-23
> **Part I**
>
> Thank you for the constructive comments and for acknowledging that **"the experimental results are comprehensive and demonstrate clear effectiveness"**, showing **"consistent improvements"** over the self-attention mechanism.
>
> ## 1. **Motivation:** How to further advance artificial neural networks in **a future where computational resources are abundant—or even redundant**?
>
> In recent years, the computational power supporting artificial neural networks has continued to increase. For example, NVIDIA has introduced the B300 GPU, which provides 288 GB of high-bandwidth memory. However, **simply deepening or widening existing networks has brought only limited performance gains.** While computational capacity can, in principle, scale indefinitely, how to enable neural networks to achieve further breakthroughs under abundant computation remains an underexplored question. Without fundamental innovation in network design, it is foreseeable that computation will soon become redundant. Therefore, we aim to explore a key question: **how can we utilize immense computation resources efficiently when they are available, and what new forms of neural networks might emerge from them?**
>
> ## 2. "Potential" to Replace Self-attention
>
> ### (1) Replace
>
> Translution preserves the same input–output interface as self-attention. It enhances the adaptive modeling capability of self-attention by introducing relative positional encoding, while remaining fully compatible with standard self-attention code implementations.
>
> ### (2) "Potential"
> As emphasized in the paper, Translution currently demonstrates only the potential to replace self-attention. Whether it can ultimately serve as a drop-in replacement requires significantly broader computational resources and much more extensive experimentation. In this work, we provide our best preliminary evidence that Translution can outperform self-attention within both ViT and GPT architectures, with particularly strong results in the ViT setting.
>
> ## 3. Comparison with CoAtNet, X-volution and Restormer
>
> Thanks for pointing out those excellent works. **We compared and cited them in the updated version.**
>
> (1) **CoAtNet:** We have compared with CoAtNet in Appendix E.5 (Relative Positional Scalar) of the original submission. Specifically, CoAtNet introduces a relative positional scalar added or appended to the self-attention mechanism, whereas our method employs offset-aware matrices to compute the Q, K, and V representations.
>
> (2) **X-volution:** X-volution adds  self-attention to convolution. Specifically, for the center element in a convolutional field, X-volution first applies a standard convolution. Second, it performs self-attention on the center element by treating it as the query. Finally, X-volution combines (adds) the two results to produce the output.
>
> (3) **Restormer:** Restormer represents more of an architectural innovation—it integrates Transformer blocks with two proposed Multi-Dconv Head Transposed Attention and Gated Dconv Feed-Forward Network  within a network framework, forming an efficient transformer for high-resolution image restoration. In contrast, Translution is more of an general operation. It can be integrated into different architectures.

---

> > ### Author Response · Authors · 2025-11-23
> > **Part II**
> >
> > ## 4. Scaling up experiments
> >
> > ### (1) Add ViT-A/16, ViT-B/16 and ViT-C/16 experiments
> >
> > Table 1.1: Accuracy (%) on the ImageNet-1K dataset with the patch size of  16. Training is
> > conducted from scratch without pretraining on external datasets, with a batch size of 256.
> >
> > | Architecture |         Method          |      #Params     |    FLOPs    |   Top-1     |    Top-5     |
> > |---------------|-----------------------|------------------|-------------|-------------|-------------|
> > | ViT-A/16       |  Self-attention        |       3.0 M         |  644.8 M   | 64.71        | 86.25         |
> > | ViT-A/16      | LoR-Translution (relative enc dim = 8) | 10.7 M | 668.6 M     | **69.28** |  **89.24** |
> > |ViT-B/16      | Self-attention | 5.7 M | 1.259 G | 73.51|  91.89 |
> > |ViT-B/16 |  LoR-Translution (relative enc dim = 8) | 21.1 M | 1.307 G | **76.20** |**93.04**|
> > |ViT-C/16 | Self-attention | 22.0 M | 4.609 G | 78.91 | 94.10 |
> > |ViT-C/16 | LoR-Translution (relative enc dim = 8) | 85.4 M | 4.800 G | **79.70** | **94.52**|
> >
> > ### (2) Add GPT-A-512 and GPT-B-512 experiments
> >
> > Table 1.2: Perplexity on OpenWebText using a batch size of 8 and a sequence length of 512.
> > | Architecture |         Method          |      #Params     |    FLOPs    |   Perplexity      |
> > |---------------|-----------------------|------------------|-------------|-------------|
> > | GPT-A-512| Self-attention | 22.1 M | 6.910 G | 47.72 |
> > | GPT-A-512| LoR-Translution (relative enc dim = 8) | 27.4 M | 6.972 G | **45.17** |
> > | GPT-B-512| Self-attention | 24.7 M | 8.879 G | 43.18 |
> > | GPT-B-512| LoR-Translution (relative enc dim = 8) | 35.5 M | 9.003 G | **39.92** |
> >
> > ### (3)  Integration with Swin-Transformer to further scale up experiment on ImageNet
> >
> > To scale up the experiments, we integrate Translution into the Swin-Transformer architecture. Since Swin-Transformer operates on local windows, it avoids the significant parameter increase typically introduced by full-image attention. Moreover, to further reduce memory consumption, we adopt a variant of LoR-Translution that applies relative encoding only to the value computation.
> >
> > Table 1.3: Accuracy (%) on the ImageNet-1K dataset of integration with Swin-Transformer.
> > | Architecture |         Method          |      #Params     |    FLOPs    |   Acc (%)
> > |---------------|-----------------------|------------------|-------------|-------------|
> > | Swin-T-Base/16       |  Self-attention        |       88 M         |  15.4 M   | 83.5        |
> > | Swin-T-Base/16       | LoR-Translution (relative enc dim = 8) | 92 M | 15.6 M     | **84.1** |
> > | Swin-T-Base/16  + ImageNet-22K Pretrain     |  Self-attention        |       88 M         |  15.4 M   | 85.2        |
> > | Swin-T-Base/16   + ImageNet-22K Pretrain    | LoR-Translution (relative enc dim = 8) | 92 M | 15.6 M     | **85.6** |
> >
> > ### (4) More Efficient Translution Variant for Scaling Up Experiments on 4096-Token OpenWebText
> >
> > To support longer-context language modeling, we design a lightweight variant of LoR-Translution. First, we apply relative positional encoding only to the value projection, reducing the computational and memory overhead. Second, we restrict the relative-position range to 64 tokens, further improving efficiency while preserving local relative structure.
> >
> > Table 1.4: Perplexity on OpenWebText using a batch size of 8 and a sequence length of 4096.
> > | Architecture |         Method          |      #Params     |    FLOPs    |   Perplexity      |
> > |---------------|-----------------------|------------------|-------------|-------------|
> > | GPT-A-4096| Self-attention | 22.8 M | 89.1 G | 160.77 |
> > | GPT-A-4096| LoR-Translution (relative enc dim = 8) | 33.4 M | 92.2 G | **157.94** |
> > | GPT-B-4096| Self-attention | 25.4 M | 137.8 G | 73.70 |
> > | GPT-B-4096| LoR-Translution (relative enc dim = 8) | 35.5 M | 149.3 G | **68.34** |
> > | GPT-C-4096| Self-attention | 61.5 M | 320.1 G | 49.40 |
> > | GPT-C-4096| LoR-Translution (relative enc dim = 8) | 85.5 M | 340.0 G | **46.44** |
> >
> > ## 5. All experiments were conducted under the **same training settings.**

---

> > > ### Author Response · Authors · 2025-11-23
> > > **Part III**
> > >
> > > ## 6. Impact of model size on performance improvement
> > >
> > > (1) We would like to emphasize that our goal is not to outperform self-attention under the same parameter or model size. Rather, our objective is to explore how further effectively advance self-attention when abundant computational resources are available. Self-attention has already proven highly effective, and further improving it is inherently challenging. Therefore, exceeding the performance of self-attention inevitably demands higher computational cost, just as self-attention itself required greater computation to outperform convolution.
> > >
> > > (2) That being said, we conduct an experiment in which self-attention and LoR-Translution are configured with the same number of parameters. To achieve this, we reduce the channel dimension of LoR-Translution accordingly. As shown in Table 1.5, Translution outperforms self-attention while using the same number of parameters and even fewer FLOPs.
> > > Table 1.5: Accuracy (%) on the ImageNet-1K dataset with the patch size of  56. Training is
> > > conducted from scratch without pretraining on external datasets, with a batch size of 256.
> > >
> > > | Architecture |         Method          |      #Params     |    FLOPs    |   Top-1     |    Top-5     |
> > > |---------------|-----------------------|------------------|-------------|-------------|-------------|
> > > | ViT-A/56       |  Self-attention        |       4.7 M         |  75.86 M   | 46.28        | 71.17         |
> > > | ViT-A/56      | LoR-Translution (relative enc dim = 8) | 4.7 M | 75.16 M     | **46.63** |  **71.50** |
> > >
> > > (3) In Sec. 4.1.3 of the article, we investigate whether the improvement of Translution stems from a larger model size or from the proposed modeling approach based on relative encoding. To this end, we replace the relative encoding in Translution with absolute encoding, which substantially increases the number of parameters. As shown in Table 5, although absolute encoding involves significantly more parameters, it achieves lower accuracy than relative encoding. Therefore, simply increasing the number of parameters does not lead to performance improvements.
> > >
> > > ## 7.  Inference efficiency: **Translution uses the same amount of computation as self-attention**
> > >
> > > Although Translution introduces a larger number of parameters, it does not incur additional computational cost compared with  self-attention. As shown in tables, Translution uses the same amount of computation as self-attention. This is because Translution simply replaces the original projection matrices $\boldsymbol{W}^q$, $\boldsymbol{W}^k$, and $\boldsymbol{W}^v$ with their offset-indexed counterparts $\boldsymbol{W}^q_{\delta_x,\delta_y}$, $\boldsymbol{W}^k_{\delta_x,\delta_y}$, and $\boldsymbol{W}^v_{\delta_x,\delta_y}$, without increasing the per-token operations.
> > > The computational complexity therefore remains unchanged.
> > > In contrast, LoR-Translution requires additional computation due to the low-rank adaptation applied during the offset-dependent projection, which increases the processing cost.
> > >
> > > However, Translution still incurs longer inference time because it requires additional GPU memory access or indexing of the offset-dependent weights $\boldsymbol{W}_{\delta_x,\delta_y}$.
> > >
> > > Table 1.6: Inference time with 224×224 input images. Inference time = Computation time +  GPU memory access time
> > >
> > > | Architecture |         Method          |      #Params     |    FLOPs    |   Computation time     |   GPU memory access time / $\boldsymbol{W}_{\delta_x,\delta_y}$ indexing time |
> > > |---------------|-----------------------|------------------|-------------|-------------|-------------|
> > > | ViT-A/56       |  Self-attention        |       4.7 M |75.86 M|      1.2 ms   |   0.1 ms  |
> > > | ViT-A/56      | LoR-Translution | 5.3 M | 77.92 M | 1.3 ms | 0.6 ms |
> > > | ViT-A/56      | Translution | 38.5 M | 75.86 M |  1.2 ms | 1.1 ms |
> > >
> > > Should you need further information, please let us know. We look forward to hearing from you soon. If you feel our efforts address your concerns, we would appreciate it if you could kindly consider raising your rating. Thank you!

---

### Author Response · Authors · 2025-11-23
**Summary**

Dear Chairs and Reviewers,

We would like to thank the reviewers for their careful and constructive comments. We also thank the reviewers for acknowledging our work is  novel (kArY, LDCV), interesting (5sr9),  strong, comprehensive or clear  effectiveness (uvaq, kArY, LDCV), offers a new perspective (5sr9) and the proposed unification maybe inspire future research (5sr9).

The paper has been revised in accordance with the reviewers’ comments and suggestions. Updates and changes are marked by green color in the revised version. Here are a few important points we would like to highlight.

### 1. Motivation: how to further advance artificial neural networks in the forthcoming era of giant computation power

In recent years, the computational power available to artificial neural networks has rapidly increased—for instance, NVIDIA’s B300 GPU now offers 288 GB of high-bandwidth memory. However, merely deepening or widening existing architectures yields diminishing performance returns. Obviously, computational resources will continue to scale. However, how neural networks should evolve to fully exploit abundant computation remains an open question. Without fundamental advances in network design, computation itself may soon become redundant.

### 2. Translution uses the same amount of computation as self-attention

Although Translution introduces a large number of parameters, it does not incur additional computational cost compared with  self-attention. As shown in tables, Translution uses the same amount of computation as self-attention. This is because Translution simply replaces the original projection matrices $\boldsymbol{W}^q$, $\boldsymbol{W}^k$, and $\boldsymbol{W}^v$ with their offset-indexed counterparts $\boldsymbol{W}^q_{\delta_x,\delta_y}$, $\boldsymbol{W}^k_{\delta_x,\delta_y}$, and $\boldsymbol{W}^v_{\delta_x,\delta_y}$, without increasing the per-token operations.
The computational complexity therefore remains unchanged.

### 3. LoR-Translution does not significantly increase the number of parameters

### 4. We should not invalidate a method solely due to its larger parameter count—after all, self-attention itself initially demanded higher computational cost before it could outperform convolution

Our goal is therefore not to surpass self-attention under fixed parameter or model-size constraints. Instead, we aim to investigate how self-attention can be further advanced when vast computational resources are available. Self-attention is already highly effective, and improving upon it is inherently difficult. Consequently, surpassing self-attention naturally requires greater computational expenditure—just as self-attention once required more computation to outperform convolution.

### 5. Translution can be regarded as an upper bound—a ceiling oriented toward future computational capabilities. While current hardware is indeed insufficient to fully support it, we can design a variety of practical variants to fit today’s computational constraints, as demonstrated by the experiments in our responses

### 6. Translution variants can outperform self-attention with the same number of parameters and less computation

### 7. Rename $\alpha$-Translution to LoR-Translution

Inspired by the comments from Reviewer LDCV, we rename α-Translution to LoR-Translution. Given the similarity in form, this new name makes the design and motivation clearer to readers.


Should you need further information, please let us know. We look forward to hearing from you soon.

Yours sincerely

Authors

---

### Meta-Review · Area_Chair_7VPt · 2026-01-05

**Summary:**

This submission was reviewed by four expert reviewers, with the ratings of: 2 borderline accept, 1 borderline reject, and 1 reject. The major concerns from the reviewers are around the number of parameters, practicality, overclaiming, differences and novelty compared to related works, the scale of the experiments wrt the claims made, and some experimental configuration concerns. After the rebuttal, some of the concerns were addressed and also acknowledged by some reviewers. But major concerns still remain not well addressed (see the section below).

Overall, there is no strong support towards a clear acceptance. After carefully considering all the reviewers' comments and the rebuttal discussions, the contributions of this paper are limited and insufficient to be a paper presented at ICLR in its current form, especially considering the outstanding concerns that are not well addressed.

**Reviewer Concerns:**

Concerns that the AC thinks were addressed by the rebuttal: the method's scaling capability; training configuration partially addressed; inference efficiency; experiment on LM lacks baselines; runtime performance.

Concerns that are still outstanding: the overclaiming and significance compared to related works; practicality; experiment scales; limited contributions and novelty when compared to related mechanisms and approaches.

**Reviewer Scores:**

According to the review comments, rebuttal, and the available discussions during the rebuttal, for each review, the reviewer might have changed their score in the way below if they had been able to participate fully in the discussion:

* Reviewer uvaq: borderline accept to borderline reject, or unchanged
* Reviewer kArY: borderline accept to accept, or unchanged
* Reviewer LDCV: borderline reject to reject, or unchanged
* Reviewer 5sr9: Reject, unchanged.

---

### Decision · Program_Chairs · 2026-01-26

Reject